# Simulating the Forest Fire Plume Dispersion, Chemistry, and Aerosol Formation Using SAM-ASP version 1.0

Chantelle R. Lonsdale[1], Matthew J. Alvarado[1], Anna L. Hodshire[2], Emily Ramnarine[2] and Jeffrey R. Pierce[2]

[1]Atmospheric and Environmental Research (AER), Lexington, MA, 02421, USA

5  [2]Department of Atmospheric Science, Colorado State University, Fort Collins, CO, 80523, USA

*Correspondence to*: Matthew J. Alvarado (malvarad@aer.com)

**Abstract.** Biomass burning is a major source of trace gases and aerosols that can ultimately impact health, air quality, and climate. Global and regional-scale three-dimensional Eulerian chemical transport models (CTMs) use estimates of the primary emissions from fires and can unphysically mix them across large-scale grid boxes, leading to incorrect estimates of the impact of biomass burning events. On the other hand, plume-scale process models allow for explicit simulation and examination of the chemical and physical transformations of trace gases and aerosols within biomass burning smoke plumes, and they may be used to develop parameterizations of this aging process for coarser grid-scale models. Here we describe the coupled SAM-ASP plume-scale process model, which consists of coupling the large-eddy simulation model, the System for Atmospheric Modelling (SAM), with the detailed gas and aerosol chemistry model, the Aerosol Simulation Program (ASP). We find that the SAM-ASP version 1.0 model is able to correctly simulate the dilution of CO in a California chaparral smoke plume, as well as the chemical loss of $NO_x$, HONO, and $NH_3$ within the plume, the formation of PAN and $O_3$, the loss of OA, and the change in the size distribution of aerosols as compared to measurements and previous single-box model results. The newly coupled model is able to capture the cross-plume vertical and horizontal concentration gradients as the fire plume evolves downwind of the emission source. The integration and evaluation of SAM-ASP version 1.0 presented here will support the development of parameterizations of near-source biomass burning chemistry that can be used to more accurately simulate biomass burning chemical and physical transformations of trace gases and aerosols within coarser grid-scale CTMs.

## 1 Introduction

Outdoor biomass burning—including wildfires, prescribed fires, and agricultural fires—is a major source of trace gases and aerosols that impact health, air quality and climate. These health- and climate-relevant primary emissions from biomass burning include species deemed as hazardous air pollutants (HAPs), such as benzene, formaldehyde, and acetaldehyde, which themselves can cause acute health effects (Wentworth et al., 2018). In addition to the pollutants directly emitted by fires, chemistry in smoke plumes can produce ozone ($O_3$), which can negatively impact human health (U.S. EPA, 2013) as well as affect vegetation, water quality, soil and the ecosystems that they support (European Environmental Agency, 2018). $O_3$ formation can occur due to the emission of nitrogen dioxide ($NO_2$), HONO, and volatile organic compounds (VOCs) and presence of sunlight (Baylon et al., 2018), with enhanced photolysis rates occurring most predominantly during midday, when photolysis rates are fastest. In 2012, the estimated median contribution of fires to maximum daily 8-hr average (MDA8) $O_3$ in Texas during the month of June was 2 ppbv, with maximum impacts of over 40 ppbv (McDonald-Buller et al., 2015). Long-range transport of fire emissions has also been found to contribute to elevated peak $O_3$ values in Europe (Ordóñez et al., 2010). Large uncertainties exist, however, in quantifying $O_3$ production, which stems from uncertainties in fire emissions, combustion efficiency, meteorological patterns, chemical and photochemical reactions, and the effects of aerosols on plume chemistry and photolysis rates. Aerosols have been shown to both

increase $O_3$ formation (e.g. scattering particles can increase photolysis rates) as well as decrease $O_3$ (absorbing aerosol and black carbon-containing aerosol can reduce photolysis rates) (Baylon et al., 2018). Hence, the aerosol composition and size distribution, which varies within and between plumes (Collier et al., 2016), and the location of the aerosol within the plume (Alvarado et al., 2015) impact $O_3$ production. The presence of clouds also impacts photolysis rates and $O_3$ production (Flynn et al., 2010). All of these factors remain highly variable and uncertain between different plumes.

Biomass burning also emits particulate matter (PM) that impacts air quality, health, and climate. PM impacts the climate directly by scattering or absorbing incoming solar radiation (e.g., Boucher et al., 2013) and indirectly by altering the properties of clouds (e.g., Pierce et al., 2007; Spracklen et al., 2011) with both effects depending on the particle size, mass and composition (Petters and Kreidenweis, 2007; Seinfeld and Pandis, 2016). Bond et al. (2013) estimated that biomass burning emits about one third of total global primary carbonaceous aerosol emissions (black carbon (BC) and organic aerosol (OA), with the size, mixing state, and chemical composition of the particles uncertain. A complex evolution of various organic trace-gas and aerosol compounds occurs as smoke ages, with all compounds containing a wide variety of volatility and reactivity that determine the ultimate partitioning into the gas- or particle-state, thus determining the size, mixing state and ultimately chemical composition of the evolving plume. Hodshire et al. (2019a) reviewed the wide-ranging results from laboratory and field studies of smoke plume aging, which show that measured net OA production/loss is dependent on the fuel and burning conditions, plume dispersion rates, and oxidant species and concentrations; however, no complete theory currently exists that can predict how OA will evolve in different plumes. Further understanding of the magnitude and extent of both the primary and secondary components of biomass burning emissions is thus required to fully understand the global impacts.

Over the last few decades, air quality regulations have resulted in a decrease in PM concentrations in the United States, as $PM_{2.5}$ is a regulated pollutant under the Clean Air Act NAAQS. McClure and Jaffe (2018), however, analyzed $PM_{2.5}$ measurements made at IMPROVE sites and found a positive trend in the 98th quantile of $PM_{2.5}$ in the Northwest and a negative trend in the rest of the US, attributing the increase to wildfires in the Northwest, similar to positive trends in MODIS AOD. They determined that wildfires are causing the increase in $PM_{2.5}$ at the 98th quantile in the Northwest which could offset anthropogenic reductions in the region. O'Dell et al. (2019) and Knorr et al. (2018) combined surface observations and satellite-based smoke plume estimates and the GEOS-Chem chemical transport model (CTM) to identify trends in summertime smoke, non-smoke, and total $PM_{2.5}$ across the US. They estimated that future PM emissions from biomass burning may exceed anthropogenic emission levels, including in densely populated areas in the eastern Europe-Russia-central Asia region. The growing relative importance of biomass burning as a source of pollution increases the need to understand in-plume chemistry and physics.

Three-dimensional (3D) Eulerian CTMs take estimates of the primary emissions from fires and unphysically mix them across large-scale grid boxes, which can lead to incorrect estimates of the ultimate impact of fires on health, air quality, and climate (e.g., Alvarado et al., 2009; Sakamoto et al., 2016; Ramnarine et al., 2019; Hodshire et al., 2019b). Thus, in order to accurately predict biomass burning effects on air quality and climate in regional and global models, a sub-grid scale representation of aged biomass-burning trace gas and aerosol evolution is required. Regarding the impact of coarse-model mixing on $O_3$, Baker et al.(2015) found that the 3D Eulerian Community Multiscale Air Quality Model (CMAQ) tended to overestimate the impact of fires on individual hourly $O_3$ measurements at US Environmental Protection Agency (EPA) Clean Air Status Trends Network (CASTNET) monitoring sites near fires by up to 40 ppbv and underestimate it further downwind by up to 20 ppbv. This behavior is consistent with an incorrect treatment of the sub-grid-scale, near-source $O_3$ and $NO_y$ chemistry, where the model underestimates the loss of

NO$_x$ (NO + NO$_2$) near the source due to formation of inorganic and organic nitrates, thus overestimating O$_3$ formation near the source. This same error leads to an underestimate of the amount of peroxy nitrates formed near the source, which then leads to an underestimate of O$_3$ formation downwind when the peroxy nitrates decompose, regenerating NO$_x$ (Alvarado et al., 2010).

Similarly, the unphysical mixing of biomass burning emissions into large-scale grid boxes can lead to incorrect estimates of OA concentrations and the aerosol size distribution (e.g., Alvarado et al., 2009; Sakamoto et al., 2016; Bian et al., 2017; Hodshire et al., 2019b; Konovalov et al., 2019). The net change of OA mass in a smoke plume as it dilutes and ages is determined from the balance between initial emissions, secondary organic aerosol (SOA) production, and evaporation of both primary organic aerosol (POA) and SOA (Bian et al., 2017; Hodshire et al., 2019b). Unphysically diluting biomass burning emissions leads to unphysical

evaporation of the POA, reduced the rates of chemical SOA formation, and more of the formed SOA remining in the gas phase in the 3D Eulerian CTMs. Similarly, the unphysical dilution reduces the aerosol number concentration, reducing coagulation rates, while the more dilute smoke will not reach the high concentrations needed to nucleate new particles. As the evolution of the aerosol size distribution in smoke plumes is primarily controlled by OA mass changes, coagulation, and nucleation, 3D Eulerian CTMs will have difficulty accurately simulation the aerosol size distribution changes without parameterizing these sub-grid scale

processes.

The initial aerosol size, number, and mass in biomass burning smoke plumes can vary with fuel type (Janhäll et al., 2010) (e.g., Boreal versus Savannah) and combustion condition (Hosseini et al., 2010) and are leading uncertainties in the predictions of PM in regional and global models (Lee et al., 2013). These inputs are often based on spatially sparse, point measurements taken at only

one stage in the atmospheric lifetime of a biomass burning plume with some measurements representing fresh emissions and some representing aged emissions (Pierce et al., 2007; Janhäll et al., 2010; Akagi et al., 2011; Hodshire et al., 2019a). These sparse inputs do not account for many of the non-linear physical and chemical changes that take place within a smoke plume near the fire, with the coarse grid scales of regional and global models (10s to 100s of kilometers) too large to resolve near-source smoke plume chemical and physical evolution. By accounting for sub-grid aerosol processes that occur in biomass burning plumes, such as

coagulation and condensation/evaporation of organic species, the biomass burning impact on aerosol number concentration and size distribution can be better simulated (Ramnarine et al., 2019). In order to resolve aerosol processes in biomass burning plumes, regional and global models thus require grid-scale-appropriate, aged aerosol emissions size distributions to accurately simulate the health and climate effects of biomass-burning aerosols in global and regional atmospheric models. Additionally, in order to better characterize the chemical processes in biomass burning plumes, improved understanding of the oxidant and radical concentrations,

photolysis rates, and parameterizations of reaction rates for different classifications of smoke is needed (Hodshire et al., 2019a).

Several types of models have been used to simulate the dispersion and transport of smoke plumes, including box models, Gaussian plume models, Lagrangian puff and particle dispersion models (e.g., CALPUFF, SCIPUFF, HYSPLIT, FLEXPART), and 3D Eulerian models (e.g., Goodrick et al., 2013 and the references therein). A smaller number of models have included the gas (e.g.,

Mason et al., 2001) and aerosol (e.g., Trentmann et al., 2003) chemistry of these plumes, and a smaller number still have tried to predict how the aerosol size distribution changes within the smoke plume (e.g., Sakamoto et al., 2016; Hodshire et al., 2019b). As an initial attempt to represent sub-grid plume chemistry and physics in coarse-grid models, Lonsdale et al. (2015) developed a parameterization of trace gas and aerosol formation in biomass burning plumes using the Aerosol Simulation Program (ASP, Alvarado et al., 2015) as a box model. ASP simulates the gas-phase, aerosol-phase, and heterogeneous chemistry of young biomass

burning smoke plumes, including the formation of O$_3$ and secondary inorganic and organic aerosol. The ASP box-model

parameterization included predicted Normalized Excess Mixing Ratios (NEMR, Akagi et al., 2011) of $O_3$, $NO_x$, PAN, and other trace gases and aerosol species in terms of the fuel type, temperature, latitude, day of year, and starting hour of fire emission. Separate parameterizations were built for each fuel type, which included Savannah, Tropical Forest, Temperate Forest, and Boreal Forest. McDonald-Buller et al. (2015) used a subset of this ASP-based parameterization to adjust the chemistry of biomass burning in the Comprehensive Air Quality Model with Extensions (CAMx) and found that this approach reduced the median impact of biomass burning on MDA8 $O_3$ in Texas by 0.3 ppbv, or 15%. However, because the parameterization was fit to the ASP box model, it did not include cross-plume gradients in trace gas and aerosol concentrations, which may be important for accurately simulating non-linear chemistry and partitioning (Garofalo et al., 2019, Hodshire et al., 2019b; Bian et al., 2017). To account for non-linear cross-plume dilution effects, Sakamoto et al. (2016) used the large-eddy simulation (LES) cloud-resolving model the System for Atmospheric Modelling (SAM; Khairoutdinov and Randall, 2003; Stevens et al., 2012), coupled with the TwO Moment Aerosol Sectional (TOMAS) microphysics module to parameterize the coagulation of aerosols in biomass burning plumes  (Sakamoto et al., 2015; 2016). This parameterization was used in Ramnarine et al. (2019) to investigate the impact of sub-grid coagulation on radiative forcing. However, while the SAM-TOMAS model used by Sakamoto et al. (2016) resolved plume gradients, their study did not include chemistry and phase partition. There remains a need for a modelling system that resolves plume gradients while simulating the chemical and physical processes relevant for air quality and climate.

To address the need for a dispersion-resolving model with online chemistry, partitioning, and microphysics that can help answer the biomass burning questions described above, we have developed an integrated model of ASP (Section 2.1) coupled with the SAM model (Section 2.2). We have evaluated the performance of the new model, SAM-ASP v1.0 described in Section 2.3, in simulating the measurements of CO, $O_3$, $NO_y$, and aerosols for the Williams Fire in California (Sections 3 and 4). This integrated model is able to simulate both the detailed chemistry, and the horizontal and vertical dispersion affecting the near-source evolution of biomass-burning gas and aerosol chemistry and physics. Model code and inputs are publicly available as described in Section 6.

## 2 SAM-ASP 2D Lagrangian Model

### 2.1 Aerosol Simulation Program v2.1

ASP (Alvarado, 2008; Alvarado and Prinn, 2009; Alvarado et al., 2009; 2015; 2016) is Fortran model that reads in the parameters for the chemical mechanism, aerosol thermodynamics, and other inputs from heavily documented ASCII files. Reading these inputs from ASCII files makes the model highly flexible. These files are read once at the beginning of the simulation and the results are stored in memory to increase computational speed. ASP v2.1 is coded as a box model with options for a plume-like configuration (with parameterized dilution) or a smog-chamber configuration, and can be called as a subroutine within larger models (e.g., Alvarado et al., 2009) when the appropriate input flags are set.

ASP uses a sectional aerosol size distribution representation (with the number of size bins adjustable at runtime) and includes modules to calculate aerosol thermodynamics, gas-to-aerosol mass transfer (condensation/evaporation), coagulation of aerosol particles, and aerosol optical properties. ASP has been extensively used to study the chemical and physical transformations of gases and particles within young biomass burning smoke plumes (less than 24 hours) (Alvarado and Prinn, 2009; Alvarado et al., 2009, 2010, 2015) and the optical properties of smoke aerosol (Alvarado and Prinn, 2009; Alvarado et al., 2009, 2015, 2016). For

example, Alvarado and Prinn (2009) used ASP v1.0 to investigate the aging of biomass burning aerosol from African savannah fires sampled during the SAFARI-2000 campaign (Hobbs et al., 2003). ASP v1.0 simulated the growth of the aerosol size distributions in this smoke plume and showed that coagulation had only a minor impact on the biomass burning aerosol growth in the first hour after emission. They also showed that the aerosol single scattering albedo increased in the first hour of aging from 0.87 to 0.90 and that the change of total aerosol light scattering with humidification decreased with aging, consistent with SAFARI-2000 studies of Magi and Hobbs (2003) and Reid et al. (2005). Alvarado et al. (2015) evaluated ASP v2.1 simulations for a fire in California (Williams fire; Akagi et al., 2012) and showed that ASP could accurately simulate most of the observed species (e.g., OA, $O_3$, $NO_x$, OH) using reasonable assumptions about the chemistry of the unidentified organic compounds. This method provides a chemically realistic way for determining the average chemistry of the thousands of organic compounds in the smoke plume, where an approach based on attempting to simulate the oxidation chemistry of each of these compounds would be computationally intractable even if all the parameters were known. The modules of the latest version of the ASP model (ASP v2.1; Alvarado et al., 2015, 2016) used in SAM-ASP v1.0 are briefly described below.

### 2.1.1 Gas-Phase Chemistry

The gas-phase chemistry within ASP v2.1 is described in detail in Alvarado et al. (2015). The chemical mechanism is integrated using a Gear-algorithm type solver. The ASP v2.1 gas phase chemical mechanism includes 1608 reactions between 621 species. All gas-phase chemistry for organic compounds containing 4 carbons or less has been "unlumped," i.e. the chemistry for each individual organic compound is explicitly resolved. This was done by following the reactions of the Leeds Master Chemical Mechanism (MCM) v3.2 (http://mcm.leeds.ac.uk/MCM/, accessed June 2012; Jenkin et al., 1997, 2003; Saunders et al., 2003; Bloss et al., 2005) for these species. The chemical mechanism of isoprene within ASP v2.1 follows the Paulot et al. (2009a,b) isoprene scheme, as implemented in GEOS-Chem and including corrections based on more recent studies (e.g., Crounse et al., 2011, 2012). The (lumped) chemistry for all other organic compounds in ASP v2.1 follows the Regional Atmospheric Chemistry Mechanism (RACM) v2 (Goliff et al., 2013).

Like most organic compounds, semi-volatile organic compounds (SVOCs) will react with OH. Most mechanisms for this chemistry (e.g., Dzepina et al., 2009) parameterize this chemistry by assuming that the SVOCs react with OH to form a lower volatility SVOC, as in the reaction:

$$\text{SVOC}_i + \text{OH} \xrightarrow{k_{OH}} \mu\text{SVOC}_{i\text{-}n}$$

(R1)

where $\mu$ is the relative mass gain due to oxidation (e.g. via O addition), $k_{OH}$ is the reaction rate with OH, and $n$ is the "volatility shift", or by how many factors of 10 to lower the $C^*$ of the product with each OH reaction. This simplified chemistry can be extended to account for the fact that the SVOCs could fragment during oxidation, leading to higher volatility products:

$$\text{SVOC}_i + \text{OH} \xrightarrow{k_{OH}} \mu(1-\alpha)\text{SVOC}_{i\text{-}n} + \mu\alpha\text{SVOC}_{i+1} + \alpha\text{VOC}_j$$

(R2)

where $\alpha$ is the fraction of $\text{SVOC}_i$ that fragments into $\text{SVOC}_{i+1}$ and $\text{VOC}_j$. Shrivastava et al. (2013) used a similar approach to show that adding SVOC fragmentation to WRF-Chem simulations of the Mexico City Plateau improved the model's ability to simulate the observed concentrations of SOA. However, the highly simplified chemistry of Reactions R1 or R2 is not appropriate for situations where reactions with the SVOC compounds are a potentially significant sink of OH, such as in a concentrated smoke

plume. Thus in ASP v2.1, the average, lumped chemistry of the SVOCs is instead parameterized in a more realistic manner for a generic organic species, following the idea of "mechanistic reactivity" (e.g., Seinfeld and Pandis, 2016). After reaction with OH SVOCs produce peroxy radicals ($RO_2$), which can react with NO to form $NO_2$ and $HO_2$, thereby regenerating OH and forming $O_3$. Reactions R3 and R4 show this more general chemical mechanism for the SVOCs:

$$SVOC_i + OH \xrightarrow{\;k_{OH}\;} RO_2 \tag{R3}$$

$$RO_{2,i} + \chi NO \xrightarrow{\;k_{RO_{2,i}}\;} \mu(1-\alpha)SVOC_{i-n} + \mu\alpha SVOC_{i+1}$$
$$+\alpha VOC_j + \beta NO_2 + \delta HO_2 \tag{R4}$$

where $k_{RO_{2,i}}$ is assumed to be $4.0\times10^{-12}$ cm$^3$ molecule$^{-1}$ s$^{-1}$ based on the reaction rate for the peroxy radicals from long-chain alkanes and alkenes with NO in RACM2 (Goliff et al., 2013). We can see that $\chi$-$\beta$ is the number of $NO_x$ lost (implicitly via the addition of a nitrate group to the product SVOCs), $1$-$\delta$ is the number of $HO_x$ lost, and $\beta$+$\delta$ is the number of $O_3$ made per reaction (by subsequent reactions of $NO_2$ and $HO_2$ to generate $O_3$). For example, the values for long-chain alkanes (HC8) in the RACM2 mechanism (Goliff et al., 2013) would be $\chi = 1$, $\delta = 0.63$, and $\beta = 0.74$, such that 0.26 $NO_x$ and 0.37 $HO_x$ are lost and 1.37 $O_3$ are

formed per reaction. Note that the mechanism of Reactions R3 and R4 is still highly simplified: we assume that reaction of SVOC with OH always produces a $RO_2$ radical, and that the $RO_2$ produced does not react with $HO_2$ or another $RO_2$. Also note that Reactions R3 and R4 represent the average chemistry of the unknown species collectively, and may not apply to any individual species in that mixture. Based on the results of Alvarado et al. (2015), we used an OH reaction rate of $1.0\times10^{-12}$ cm$^3$ molecule$^{-1}$ s$^{-1}$ for Reaction R3, and values of $\mu = 1.075$, $\alpha = 0.5$, $n = 1$ $\chi = 1$, $\delta = 0.6$, and $\beta = 0.5$ as the defaults in ASP v2.1.

### *2.1.2 Aerosol Size Distribution, Thermodynamics and Gas-Particle Mass Transfer*

The aerosol size distribution in ASP is represented using a moving-center sectional approach (Jacobson, 2002). The current ASP SOA formation module is the semi-empirical Volatility Basis Set (VBS) model of Robinson et al. (2007) linked to the RACM2 chemical mechanism following the approach of Ahmadov et al. (2012), with the semi-volatile and intermediate-volatility organic compound (S/IVOC) chemistry expanded and optimized for biomass burning following the results of Alvarado et al. (2015), with

the saturation concentration, C*, ranging from $1.0\times10^{-2}$ to $1.0\times10^{6}$ ug m$^{-3}$ at 300 K with 9 bins total.

Equilibrium concentrations both within the aerosol phase and between the gas and aerosol phase are calculated using the Mass Flux Iteration (MFI) method to solve for the gas- and aerosol-phase concentrations at equilibrium for a given reaction (Section 17.11 of Jacobson, 2005). Mass transfer between the gas and aerosol phases is calculated in ASP using a hybrid scheme, where

the condensation of $H_2SO_4$ follows the flux-limited condensation equations while the kinetic condensation/evaporation of organic species are calculated using a Gear algorithm (due to the stiff nature or kinetic OA partitioning across volatilities and particle sizes). However, $NH_3$, $HNO_3$, and HCl are assumed to be in equilibrium (Alvarado and Prinn, 2009). Aerosol coagulation is calculated using a semi-implicit scheme (Jacobson, 2005) with a Brownian coagulation kernel.

### *2.1.3 Aerosol Optical Properties*

ASP v2.1 (Alvarado et al., 2015, 2016) uses spectrally varying complex refractive indices for all aerosol components based on Hess et al. (1998). The refractive index of the inorganic aqueous solution (if present) is calculated using the molar refraction approach of Tang (1997). ASP v2.1 includes four BC mixing-rule options for the calculating absorption and scattering coefficients:

(1) a volume-average dielectric constant mixing rule with BC internally mixed with other species; (2) a core-shell mixing rule, where a spherical, internally mixed BC core is surrounded by a spherical shell of all other aerosol components; (3) the Maxwell Garnett mixing rule (Maxwell Garnett, 1904) with BC internally mixed with other species; and (4) an external mixture of BC and the other aerosol components. Mie calculations of aerosol optical properties for each bin of the size distribution are performed within ASP using the publicly available program DMiLay, which is based on the work of Toon and Ackerman (1981). Only the

core-shell parameters were used in this study.

## 2.2 SAM

The SAM v6.10.10 model is a Fortran code has been used to study aerosol-cloud-precipitation interactions in stratiform and convective clouds (Ovchinnikov et al., 2014; Fan et al., 2009). The standard SAM model (Khairoutdinov and Randall, 2003, http://rossby.msrc.sunysb.edu/~marat/SAM.html) includes different options of detailed cloud microphysics, as well as coupled

radiation and land-surface models. SAM is able to resolve boundary layer eddies, while parameterizing smaller-scale turbulence and microphysics for the LES (vs cloud-resolving) model option. The dynamical framework of the model is based on the large eddy simulation (LES) model of Khairoutdinov and Kogan (1999). Besides using the anelastic equations of motion in place of the Boussinesq equations of the LES version, SAM uses a different set of prognostic thermodynamic variables and employs a different microphysics scheme. The computer code was designed to run efficiently on parallel computers using the Message Passing

Interface (MPI) protocol. The detailed description of the model equations is given in the appendix A of Khairoutdinov and Randall (2003).

The prognostic thermodynamical variables of the model are the liquid water/ice moist static energy, total nonprecipitating water (vapor + cloud water + cloud ice), and total precipitating water (rain + snow + graupel). The liquid water/ice moist static energy

is, by definition, conserved during the moist adiabatic processes including the freezing/melting of precipitation. The cloud condensate (cloud water + cloud ice) is diagnosed using the so-called "all-or-nothing" approach, so that no supersaturation of water vapor is allowed. Despite being called a nonprecipitating water substance, the cloud ice is actually allowed to have a nonnegligible terminal velocity. The partitioning of the diagnosed cloud condensate and the total precipitating water into the hydrometeor mixing ratios is done on every time step as a function of temperature. The diagnosed hydrometeor mixing ratios are then used to compute

the water sedimentation and hydrometeor conversion rates.

The finite-difference representation of the model equations uses a fully staggered Arakawa C-type grid with stretched vertical and uniform horizontal grids. The advection of momentum is computed with the second-order finite differences in the flux form with kinetic energy conservation. The equations of motion are integrated using the third-order Adams–Bashforth scheme with a variable

time step. All prognostic scalars, including the chemical tracers of ASP v2.1, are advected using a fully three-dimensional positive definite and monotonic scheme of Smolarkiewicz and Grabowski (1990). The subgrid-scale model employs the so-called 1.5-order closure based on a prognostic subgrid-scale turbulent kinetic energy. The model uses periodic lateral boundaries, and a rigid lid at the top of the domain. To reduce gravity wave reflection and buildup, the Newtonian damping is applied to all prognostic variables

in the upper third of the model domain. The surface fluxes are computed using Monin–Obukhov similarity. SAM can be driven by reanalysis data that includes large-scale forcings, initial sounding profile, radiation heating rates, and surface fluxes. SAM has the ability to add a large amount of modeled tracer species to the cloud resolving model simulation but does not contain aerosol and chemistry packages.

The SAM model is flexible with different choices for advection scheme, turbulence parameterization, radiation, and cloud microphysics. The configuration used in SAM-ASP v2.1 includes the use of a positive definite monotonic advection scheme with a non-oscillatory option, the 1.5-order TKE closure for sub-grid scale turbulence, the microphysics scheme of Morrison et al. (2005), and the CAM radiation code.

**2.3 Model Coupling**

We coupled ASP v2.1 to the SAM v6.10.10 model to resolve dispersing biomass burning plumes with detailed chemistry and aerosol physics. The resulting Fortran code uses all of the same numerical solvers as the individual ASP and SAM models, which are discussed above. The SAM model has been previously been coupled with the TOMAS microphysics module to reproduce observed dispersion and new particle formation in coal-fired power-plant plumes (Lonsdale et al., 2012; Stevens et al., 2012) and to study the coagulation of aerosols in biomass burning plumes (Sakamoto et al., 2016). The coupling of SAM-ASP v1.0 was performed similar to the coupling of SAM and TOMAS described in Stevens et al. (2012), and the coupling of ASP to the Cloud Resolving Model (CRM6) described in Alvarado et al. (2009). SAM was updated to transport over 600 gas-phase chemical species calculated in ASP and the 840 aerosol parameters (number concentrations for each bin and mass concentrations for each aerosol species in each bin) and to simulate the emission of the fire smoke by making substantial changes to the tracers.f90 subroutine of SAM. While the number of chemical species and number of size bins is flexible in ASP v2.1 and read in from ASCII input files, these values are hard-coded into the coupled SAM-ASP v1.0 model. There is no coupling of the ASP aerosols with the SAM cloud microphysics scheme in SAM-ASP v1.0.

The tracers.f90 subroutine of SAM was also modified to communicate the solar zenith angle and initialize gas and aerosol tracer concentrations based on SAM meteorological parameters. The coupling takes place via a new ASP subroutine called within tracers.90 in SAM, called SAM_wrapper, which collects the current gas and aerosol concentrations and other parameters and passes them into ASP via the routines in ASP/StepASP.f90. StepASP.f90 performs unit conversions, passes the information into the ASP v2.1 box model, and then calculates the gas-phase chemistry (including heterogeneous chemistry), aerosol thermodynamics, and aerosol coagulation using the routines of ASP v2.1 described in Section 2.1, which are documented in Alvarado (2008) and Alvarado et al. (2015).

In this project, SAM was configured as a moving, 2D Lagrangian wall oriented normal to the mean wind direction in the layer of smoke injection (between 1200 and 1360 m in our example case shown here) as in Figure 1, reproduced from Sakamoto et al. (2016). Note that wind shear in the meteorological dataset used for boundary conditions also impacts the coupled model – the downwind (x) direction is determined once and from then on the dynamics occur in this 2D plane based on the boundary condition forcing and the model advection and turbulence schemes. Stevens and Pierce (2014) showed that this 2D model configuration does well in simulating $SO_2$ and $NO_x$ dispersion in power-plant plumes as compared to airborne measurements.

Photolysis rates are calculated in ASP using offline lookup tables generated by the Tropospheric Ultraviolet and Visible (TUV) radiation model (Bais et al., 2003) that depend on solar zenith angle and overhead $O_3$ columns. SAM-ASP v1.0 does not currently account for the impact of aerosols on these photolysis rates. ASP is run as a subroutine in each SAM master time step (10 seconds for the simulations here). The SAM model handles all tracer transport and supplies the temperature, pressure, air density, solar zenith angle, mass emissions flux, and initial gas concentrations to ASP, while ASP calculates the gas and aerosol processes within each grid box. SAM-ASP v1.0 currently does not calculate deposition but may be added in the future (the plume does not contact the ground for the case described in this paper). The grid boxes in the 2D moving wall have a 500 m x 500 m horizontal resolution with a 120 km total domain width (and 500 m in the with-wind direction, 1 box) and 40 m vertical resolution with a total vertical extent of 3 km. The simulation here was spun up for 1800 s prior to emissions following Stevens and Pierce (2014). The resolution and time steps described here are flexible and should be customized depending on plume and meteorological characteristics.

When ASP v2.1 is run as a Lagrangian box model, it needs the initial concentrations within the plume to be specified. However, as SAM-ASP v1.0 can simulate the dispersion of the smoke horizontally and vertically, we added the capability to calculate the initial concentrations based on the mass emissions flux ($M$, kg burned m$^{-2}$ s$^{-1}$), emission factors ($EF$, g (kg burned)$^{-1}$), and fire area ($A$, m$^2$ and assumes a square shape) for biomass burning species (Akagi et al., 2011; Sakamoto et al., 2015). The formula is:

$$\Delta m_q = M \cdot EF_q \cdot A \cdot \Delta t / BM \qquad (1)$$

Where $\Delta m_q$ is the mass mixing ratio (kg q/kg air) of species q, which are the units used in SAM for tracer species, and $BM$ is the mass of air in the emission box (in kg). This allows SAM-ASP v1.0 to better represent a wide range of fire sizes and intensities. To reduce computation time, ASP is only called in the boxes that are impacted by smoke in each SAM timestep, defined as any grid box having a concentration of CO greater than a user-defined threshold (based on background concentrations determined by ambient fire measurements, here 150 ppb). The coupled SAM-ASP v1.0 model was run on 12 processors with 4 GB each, which should be considered the minimum system requirements.

## 3 SAM-ASP Simulations of the Williams Chaparral Fire

We evaluated the performance of the newly coupled SAM-ASP v1.0 model by comparing model output to observations of the Williams Fire made by Akagi et al. (2012), which was previously simulated using ASP in a Lagrangian box model by Alvarado et al. (2015). Emission ratios for this simulation were based on observed relative background-corrected concentration close to the source from Alvarado et al. (2015) and included observed values for many gas-phase species measured by Akagi et al. (2012). Plume injection height was set between 1200 to 1400 m, as this was the height at which the plume was observed to level off, where a small amount of vertical mixing can be seen as the plume ages.

The large-scale meteorological forcing in SAM-ASP v1.0 is driven by the 3-hour, 32 km resolution North American Regional Reanalysis (NARR; Mesinger et al., 2006) meteorology dataset. The fire simulated in this study to evaluate the SAM-ASP v1.0 model was a prescribed fire measured on 17 November 2009 called the Williams Fire (Akagi et al., 2012). This fire covered 81 hectares north of Buellton, CA with the fuel type classified as chaparral and the vegetation burned consisting of coastal sage scrub and scrub oak woodland understory. Surface temperatures ranged from 19 degrees C at 09:00 local time to 24 degrees C at 12:20 local time with clear skies throughout the fire duration. The plume built up gradually during the day with most of the smoke rising

to ∼1200–1336 m above mean sea level and then drifting in a northeast direction. Two flights were conducted during the day on board a US Forest Services Twin Otter aircraft to sample initial emissions and aged smoke with an Airborne Fourier Transform Infrared Spectrometer instrument from the University of Montana taken during both flights including background measurements sampled at similar altitudes to in-plume measurements just outside the plume. Trace gas species emission factors determined as

initial emissions (within minutes of the emission source) by Akagi et al. (2012) are used to initialize ASP and included; $O_3$, nitrous Acid (HONO), ammonia ($NH_3$), ammonium ($NH_4$), nitric oxide (NO), $NO_2$, $NO_x$ (as NO), OA and peroxy acetyl nitrate (PAN). Additional emission factors not measured, but needed to initialize ASP, were included using emission factors from Table 2 of Akagi et al. (2011), with the full list of ASP species provided in the model code repository described in Section 6. Initial aerosol size distribution information is inferred from the smoke study of Grieshop et al. (2009a, b). Full details of the fire and measurements

are also further described in Akagi et al. (2012). Model input background concentrations were assumed based on measurements taken outside of the defined plume. Additional static inputs required by ASP include a photolysis rate parameterization based on the time and latitude of the fire, and chemical data on aqueous phase ions and inorganics. Details of the static ASP inputs are further described in Alvarado (2008).

The model used here has 10 size bins with the total number of fire-emitted particles derived from multiplying the CO flux (based on measured values) by the ratio of particle number enhancement (number of particles $cm^{-3}$) to CO enhancement (ppb) ($\Delta N/\Delta CO$ = 23.7 particles $cm^{-3}$ $ppb^{-1}$). We use a number mean diameter of 0.1 um, and a standard deviation of 1.9 based on the wood smoke study of Grieshop et al. (2009a, b).

The NEMR calculations were determined by calculating the average species of interest (X) and CO concentration across the plume, as was done in the measurements. The NEMR ($\Delta X/\Delta CO$ or $\Delta X/\Delta CO_2$) is then calculated relative to CO or $CO_2$ since they have relatively long lifetimes for the fire location, low background variability, and there were no other major nearby sources as described in Akagi et al. (2012):

$$\Delta X/\Delta CO = (X_{\text{in-plume}} - X_{\text{background}}) / (CO_{\text{in-plume}} - CO_{\text{background}}) \quad\quad (2)$$

Note that in general, NEMRs are imprecise indicators of chemical changes, especially for plumes that have traveled far from their original source and may have mixed with different types of background air and thus defining a single background concentration to subtract from the plume concentration is not a realistic approach (e.g., Yokelson et al., 2013). However, for the Williams fire, the excess mixing ratios downwind tended to vary slowly in time and space compared to measurement frequency and the background

value was computed from the average of a large number of points at the plume altitude (but outside the plume, Akagi et al., 2012).

Averaged NEMR values over the full horizontal domain of the model for the vertical level with the peak $\Delta CO$ (and $\Delta CO_2$) can be used to compare with aircraft observations of biomass burning plumes. Figure 2 shows a horizontal slice of the simulated $\Delta CO$ concentrations (ppbv) as the plume moves downwind (birds eye view at 1200 m above ground). Note the initial plume was

distributed across two horizontal gird boxes (initial plume width of 1 km) and four vertical grid boxes (initial height from 1200 m to 1360 m) and was rectangular. The emission were distributed proportional to the density of air in each grid box, and initially propagated downward due to wind shear and diffusion. The dilution of the plume can be seen in the $\Delta CO$ values as high as 16,000 ppbv between the center of the plume within the first hour after initial emission to five hours downwind, where the plume was modeled to be approximately 100 km wide, with an average in-plume $\Delta CO$ concentration of approximately 1000 ppbv. Figure 3

shows a vertical slice of (looking into) the plume at 1, 2, and 5 hours downwind for $\Delta CO$, $\Delta O_3/\Delta CO$ and $\Delta OA/\Delta CO_2$, and these

results will be discussed in the following sections. Note that $\Delta CO_2$ was used to as the NEMR denominator for OA, as in Akagi et al. (2012) and Alvarado et al. (2015), as in the field study OA and $CO_2$ were measured on the same inlet while CO was measured on a different inlet. The uncertainty in the Lagrangian age (horizontal error bars in Figure 3) was calculated as in Akagi et al. (2012), where the one-sigma uncertainty in the average horizontal wind speeds during the sampling period were propagated through

the plume age calculation, assuming the distance calculation was accurate.

For better comparison between the ASP v2.1 box model of Alvarado et al. (2015) and SAM-ASP v1.0, all emission ratios and background concentrations were made identical in box models. The same gas-phase chemical mechanism, aerosol thermodynamics routines and parameters, aerosol size distribution routines and parameters, and other chemical parameters were used. Thus the key

difference between the two models is the treatment of plume dilution and mixing (with minor differences due to vertical temperature, pressure, and humidity variations in SAM-ASP v1.0 versus constant parameters used in ASP v2.1). In ASP v2.1, the plume is a single well-mixed box and dilution is parameterized by assuming a one-way addition of background air to the plume. As in Mason et al. (2001) we assume a Lagrangian parcel of fixed height ($H$) and length, but variable width $y(t)$. This assumes the plume is well-mixed vertically and capped at the top and bottom by a strong stable layer (or the surface). The temperature and

pressure of the parcel are assumed to be constant. The effect of plume dispersion on concentrations is then (Mason et al., 2001; Alvarado, 2008):

$$\left(\frac{dC_q}{dt}\right)_{disp} = -\frac{1}{y(t)}\frac{dy(t)}{dt}\left(C_q - C_q^a\right) \qquad (3)$$

where $C_q$ is the concentration of species $q$ within the parcel (molecules/cm$^3$) and $C_q^a$ is the concentration of species $q$ in the atmosphere outside of the parcel. The form of $y(t)$ is assumed to be $y(t) = y_o^2 + 8K_y t^2$, where $y_o$ is the initial plume width (Mason

et al., 2001). $K_y$ represents the horizontal diffusivity of the atmosphere. The effect of plume dispersion then becomes

$$\left(\frac{dC_q}{dt}\right)_{disp} = -\frac{4K_y}{(y_o^2 - 8K_y t^2)}\left(C_q - C_q^a\right) \qquad (4)$$

This equation is used with the observations of the rate of change of excess CO in the Williams fire plume to derive best fit values for $K_y$ using the observed value of $y_o$.

In SAM-ASP v1.0, horizontal and vertical mixing between the boxes of the Lagrangian wall are calculated as part of the tracer transport routines of SAM 6.10.10 described in Section 2.2 (Khairoutdinov and Randall, 2003). In addition, unlike the ASP v2.1 box model of Alvarado et al. (2015), plume gradients are preserved in SAM-ASP v1.0. Thus, the chemistry taking place in the center of the plume may differ from that in the edges of the plume, potentially changing the plume average NEMRs from those calculated with the well-mixed box assumption in ASP v2.1.

**3.1 Gas-phase Simulations**

The in-plume CO enhancement ($\Delta CO = CO_{in-plume} - CO_{background}$, in ppbv) and NEMRs (Eq. 1) for $O_3$, are shown in Figures 3 and 4. NEMRs for PAN, $NO_x$, HONO and $NH_3$ are shown in just Figure 4, where the average NEMR across SAM-ASP v1.0 grid boxes is calculated where the CO concentrations are above a background threshold of 150 ppbv (based on measurements). In Figure 4, horizontal error bars indicate the age uncertainty of the measurements, with a best estimate of the starting NEMR and

uncertainty discussed in Akagi et al. (2012), which uses a slope-based fire-average ER as a best estimate of the likely starting NEMR for primary species measured in individual smoke transects. The SAM-ASP v1.0 model was qualitatively able to simulate the dilution of CO in the smoke plume after 2 hours to within the uncertainties of the measurements but with an underestimate of dispersion in the first two hours. As ASP v2.1 currently uses a fixed function to simulate dilution, we were unable to test how using

the SAM-ASP predicted dilution of CO to ASP v2.1 would alter the box model results. SAM-ASP v1.0 also correctly simulated the chemical loss of $NO_x$ and HONO and formation of PAN and $O_3$ within the plume. After two hours of model simulated $O_3$ (second row in Figure 3), it can be seen that the edges of the plume have higher concentrations than the center, a feature that cannot be represented in a box model simulation. This $O_3$ enhancement at the edges may be a result of less NO titration at the plume edges.

We expect larger $O_3$ edge effects in future work when the TUV radiation model is coupled online and interacts with plume aerosols. $N_{H}3$ concentrations in the model were overestimated (model value of $\Delta NH_3/\Delta CO$ of 0.04 ppb ppb$^{-1}$ at 5 h, rather than the measured value of 0.02 mol mol$^{-1}$). This is in contrast to the results of the ASP Lagrangian box model study of Alvarado et al. (2015), where the box model simulated $NH_3$ closer to measured values at all hours downwind. The results for this gas are very sensitive to the amount of sulfate and nitrate formed in the plume, the dilution of the plume as it affects the volatilization of $NH_3$ from the aerosol,

as well as the relative humidity and temperature, all of which slightly differ between ASP v2.1 and SAM-ASP v1.0, but we have not yet determined which difference is driving the ammonia discrepancy. The lack of vertical variation in the SAM-ASP plume in Figure 3 may be due to the use of photolysis rates that are not altered by the simulated aerosol scattering and absorption in this version of SAM-ASP. Thus, while the photolysis rates vary with time, they do not vary horizontally or vertically, with future work needed to incorporate in-line, vertically varying photolysis consideration.

**3.2 Aerosol Simulations**

OA NEMR results from the ASP box model (Alvarado et al., 2015) and measurements from Akagi et al. (2012) are compared to cross-plume-averaged SAM-ASP output in Figure 5. In general, the SAM-ASP results show slightly slower initial dilution than the box model, with the initial increase in OA due to the 2D wall staying over the emission area, thus evaporative driven decreases have not dominated yet. This difference in dilution rate, and thus OA NEMR, is due to dilution in the box being forced to match

measurements while in SAM-ASP, the meteorology and initial plume width determine the relative dispersion rate (Alvarado et al., 2015). Within the first hour after emission, SAM-ASP has less dilution than the box model (Figure 4a and 5), leading to a higher OA concentration, which in turn leads to less evaporation of OA to intermediate and semi-volatile vapors, explaining the larger OA NEMR for this initial time period. However, SAM-ASP has greater dilution than the box model after 2 hours (though both falling within measurement uncertainties in Figure 4a), which lead to more OA evaporation in SAM-ASP than in the box model,

leading to a lower OA NEMR after 2 hours, better matching the measurements. We note, however, that there are considerable uncertainties in the volatility distribution of the simulated POA as well as the SOA chemistry, so there may be multiple ways to improve modeled OA NEMR. The bottom panels of Figure 3 shows that the OA NEMR in the model initially decreases faster than the core, driven by dilution. However, after several hours the OA NEMR at the edges increases, showing that SOA production in those locations is exceeding evaporation in those locations. Thus, in both models the initial POA partially evaporates, but this is

balanced by oxidation of the S/IVOCs in the gas-phase, which then condense as SOA. This initial evaporation followed by net SOA production is consistent with the theoretical studies of Bian et al. (2017) and Hodshire et al. (2019b); however, those studies did not explore this behavior in the plume edges versus the core. SAM-ASP will be used in future work to investigate these plume edge versus core differences within field observations.

We also compared the predictions for aerosol size distribution changes between the two models (Figure 6). Note that as no size distribution measurements were taken for this fire, we cannot compare these simulations with observations. The top plot shows the average size distribution of the background air in the SAM-ASP simulation. We again average the SAM-ASP results across grid boxes where CO concentration are above the CO threshold (150 ppbv) in each timestep. Both models suggest that this fire showed

little net aerosol diameter growth (bottom two plots), as shrinking due to evaporative losses driven by dilution compensates growth by coagulation and the oxidation (and reduction in volatility) of the organic vapors, consistent with the OA NEMR results above.

**4 Conclusions**

We have described a new coupled model, SAM-ASP v1.0, for simulating the gas and aerosol chemistry within biomass burning smoke plumes. The model adds the Aerosol Simulation Program v2.1 (ASP v2.1) as an embedded subroutine within the System for Atmospheric Modeling v6.10 (SAM v6.10). When configured as a 2D Lagrangian wall, the newly coupled SAM-ASP model allows for a detailed examination of the chemical and physical evolution of fine-scale biomass burning plumes.

SAM-ASP is able to simulate the complex, non-linear production of $O_3$ and changes in PM as plumes age. It is able to resolve the cross-plume chemistry, gas-to-particle partitioning, and microphysics that coarser grid-scale CTMs are not able to. Model results indicate SAM-ASP is able to accurately simulate the dilution of CO in a California chaparral smoke plume mostly, except for a slight initial underprediction, as well as accurately predict the chemical loss of NOx and HONO and production of $O_3$ and PAN within the plume. SAM-ASP also resolves the cross-plume concentration of trace-gases and aerosol. However, when compared to observations, the simulation with SAM-ASP did not show any significant differences with respect to a much simpler box model simulation, potentially because the photolysis rates within both simulations were identical, rather than allowing the photolysis rates to vary with predicted aerosol concentrations.

Future work will involve testing SAM-ASP simulations against observed plume crosswind and vertical gradients as well as size distributions. Future work will also include the development of a biomass burning parameterization of plume-scale chemical and physical trace gas and aerosol evolution for use in coarser grid-scale CTMs (that cannot resolve plumes) as well as the implementation of on-line photolysis calculations to explicitly simulate the effect of in-plume aerosols on photolysis rates.

**5 Data Availability**

SAM-ASP 1.0 source code is available for download through the SAM website at http://rossby.msrc.sunysb.edu/~marat/SAM.html through request to the SAM model developer, Dr. Marat Khairoutdinov. Separate ASP model code, model inputs, outputs and post-processing steps described in this study are available in a public repository at https://doi.org/10.5281/zenodo.3363995, doi: 10.5281/zenodo.3363995.

*Competing Interests.* The authors declare they have no conflicts of interest.

*Author Contribution.* C.R. Lonsdale provided the design and performed the execution of the model implementation, simulation and evaluation of the SAM-ASP code and prepared this manuscript. M. J. Alvarado and J. R. Pierce provided oversight and leadership of the overall development and acquired the financial support for the project leading to this publication, as well as contributed to the review and editing of this manuscript. A. L. Hodshire and E. Ramnarine provided verification of the overall reproducibility of model results, presented complementary published work and contributed to the review and editing of this manuscript.

*Acknowledgements.* This work was supported by NOAA Atmospheric Chemistry, Carbon Cycle and Climate Program awards #NA17OAR4310001, #NA17OAR4310002 and #NA17OAR4310009, as well as NSF Atmospheric Chemistry Program Grants #1559598 and #1559607 as well as by the State of Texas through the Air Quality Research Program administered by The University of Texas at Austin by means of a Grant from the Texas Commission on Environmental Quality.

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

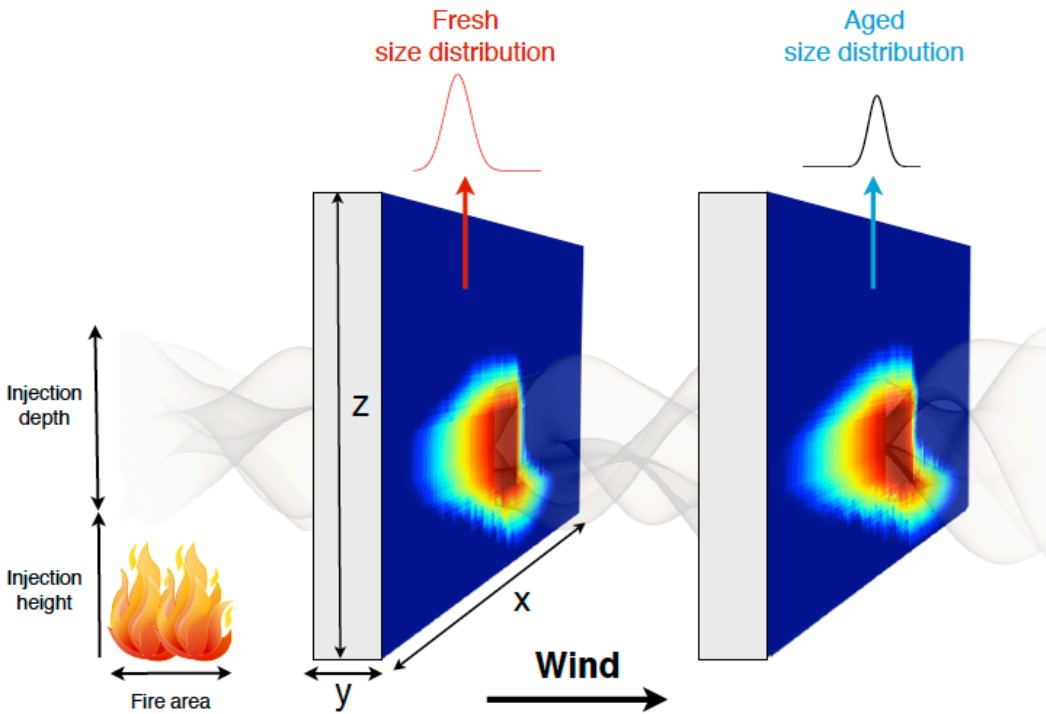

**Figure 1. Schematic of the 2D Lagrangian Wall configuration of SAM-TOMAS and SAM-ASP v1.0. Reproduced from Sakamoto et al. (2016).**

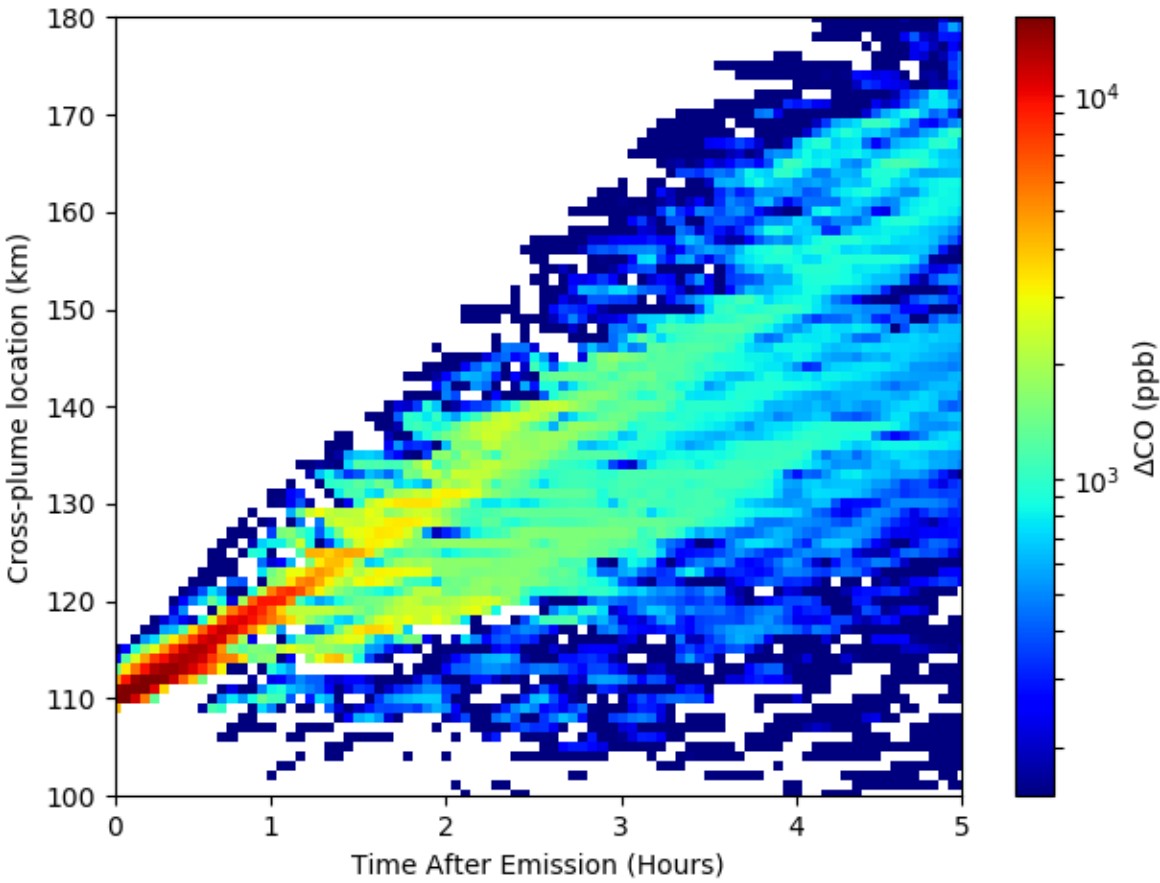

**Figure 2. SAM-ASP Williams fire simulation of cross-plume location versus time since emission at a vertical height of 1200 m for ΔCO. Note that figure is zoomed in on the plume with white background indicating a concentration less than 150 ppb.**

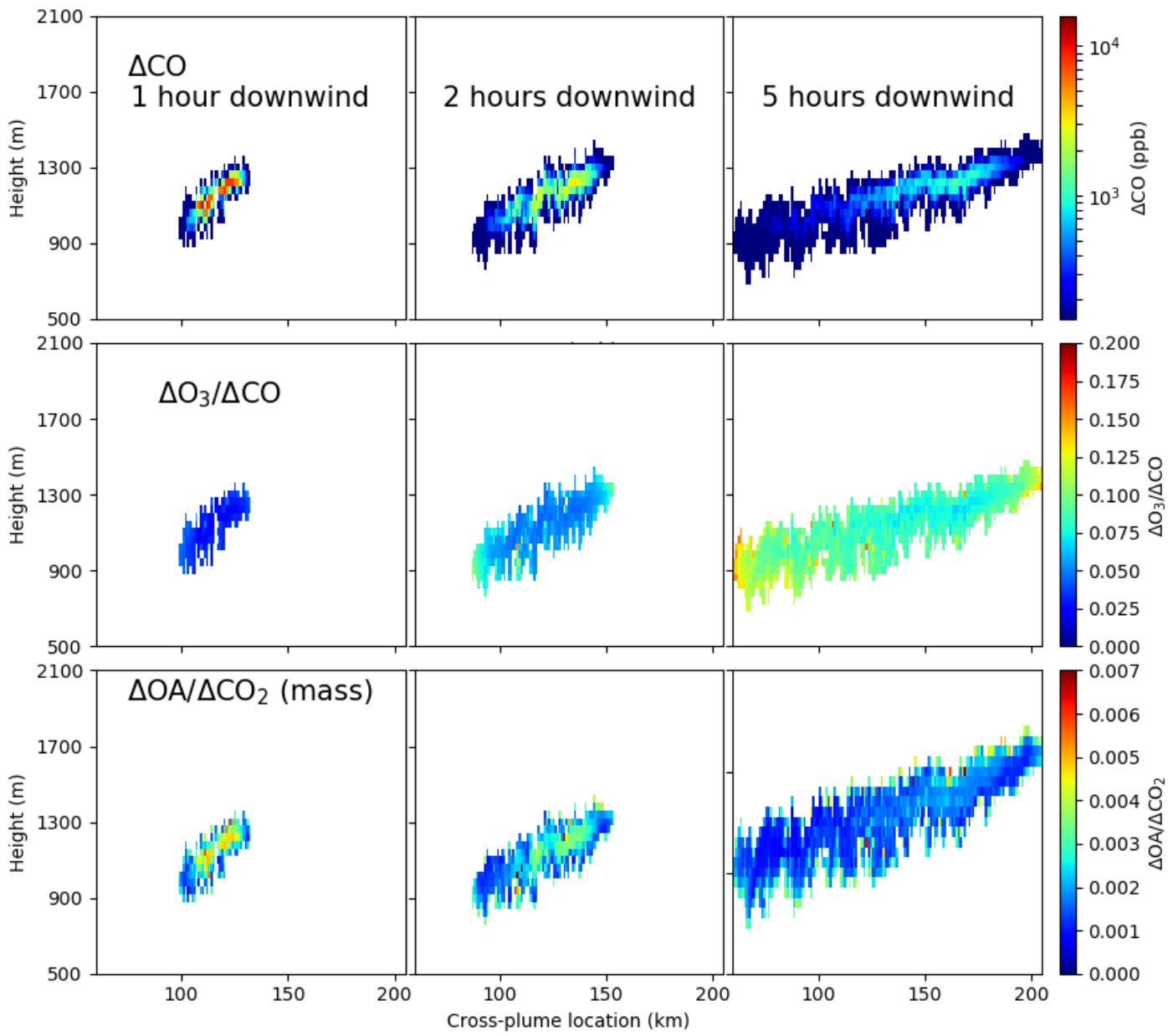

**Figure 3. Height vs cross-plume location at 1 hour (left column), 2 hours (center column) and 5 hours (right column) downwind of fire source for ΔCO (top row), ΔO₃/ΔCO (ppb/ppb, center row) and ΔOA/ΔCO₂ (bottom row). Note that figure is zoomed in on the plume and white indicates a CO concentration less than 150 ppb.**

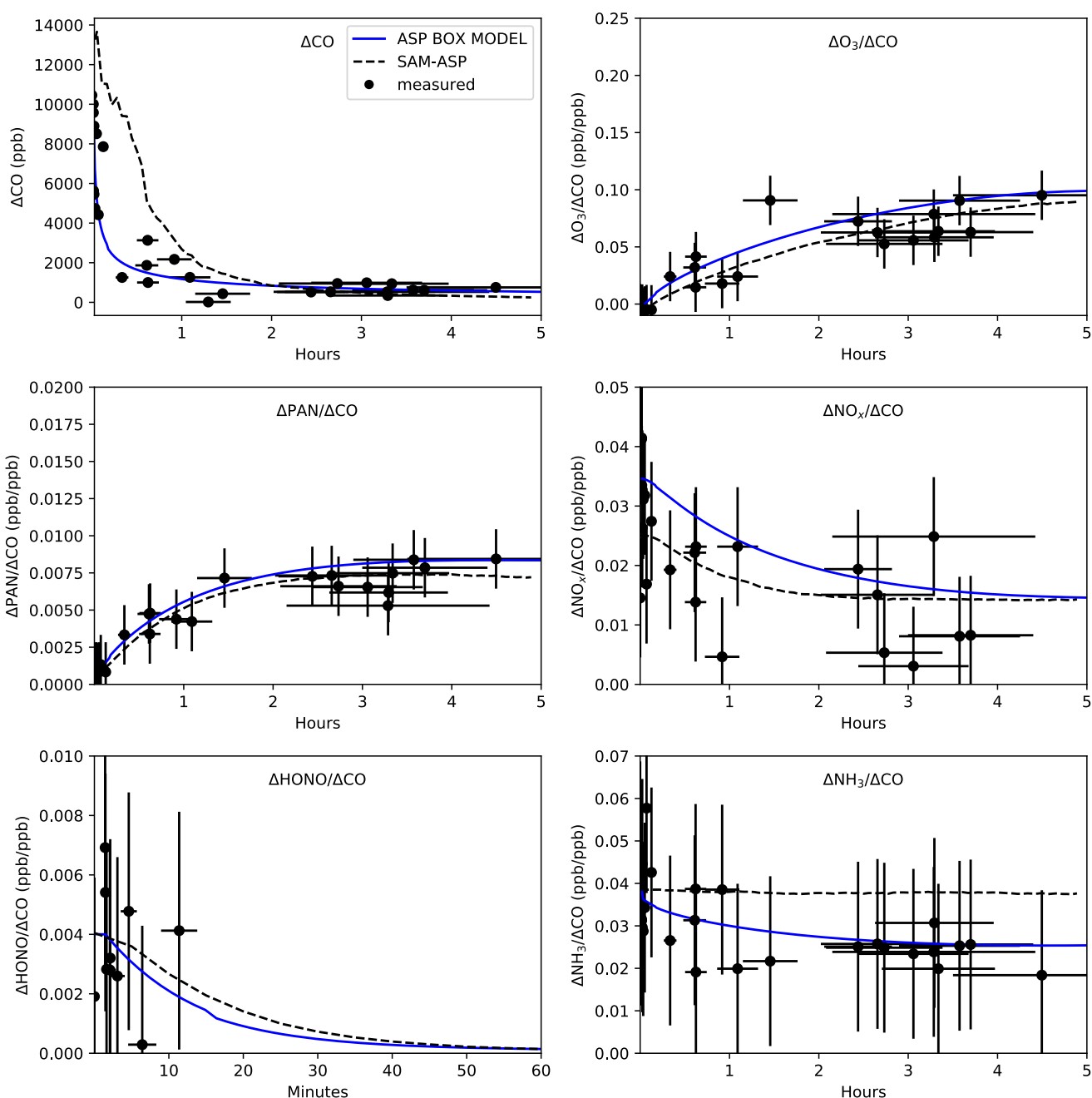

**Figure 4. Cross-plume averaged ΔCO and O₃, PAN, NOₓ, HONO and NH₃ NEMRs (ΔX/ΔCO) as a function of plume age for the ASP box model (solid line, reproduced from Alvarado et al., 2015) and SAM-ASP model (dashed-line) results compared to measurements from Akagi et al. (2012) (dots). The horizontal error bars indicate the age uncertainty of the measurements while the vertical errors bars are the uncertainty of the measured value.**

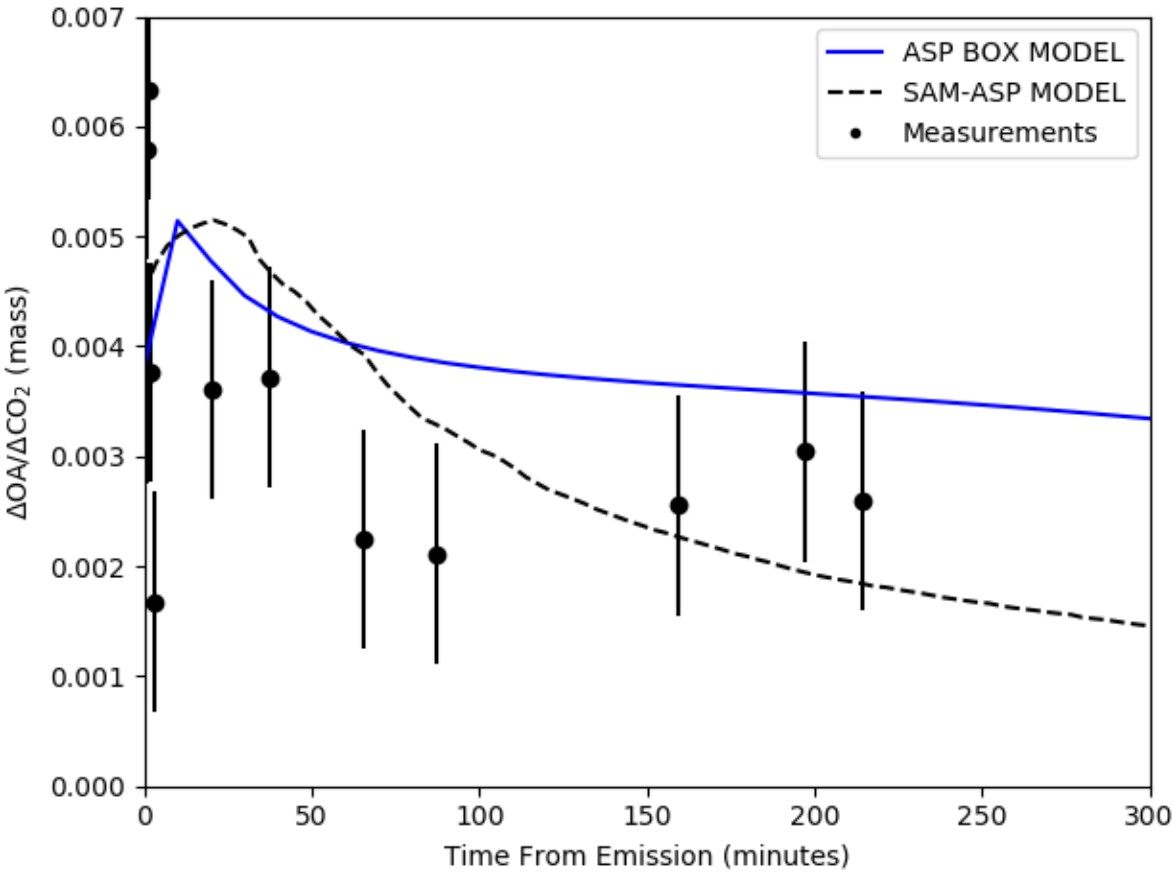

**Figure 5. Cross-plume averaged OA NEMR (ΔOA/ΔCO$_2$) for the Williams fire from SAM-ASP simulations (dashed line), the ASP box model results as described in Alvarado et al. (2015) (solid lines), and OA measurements (back dots) described in Akagi et al. (2012).**

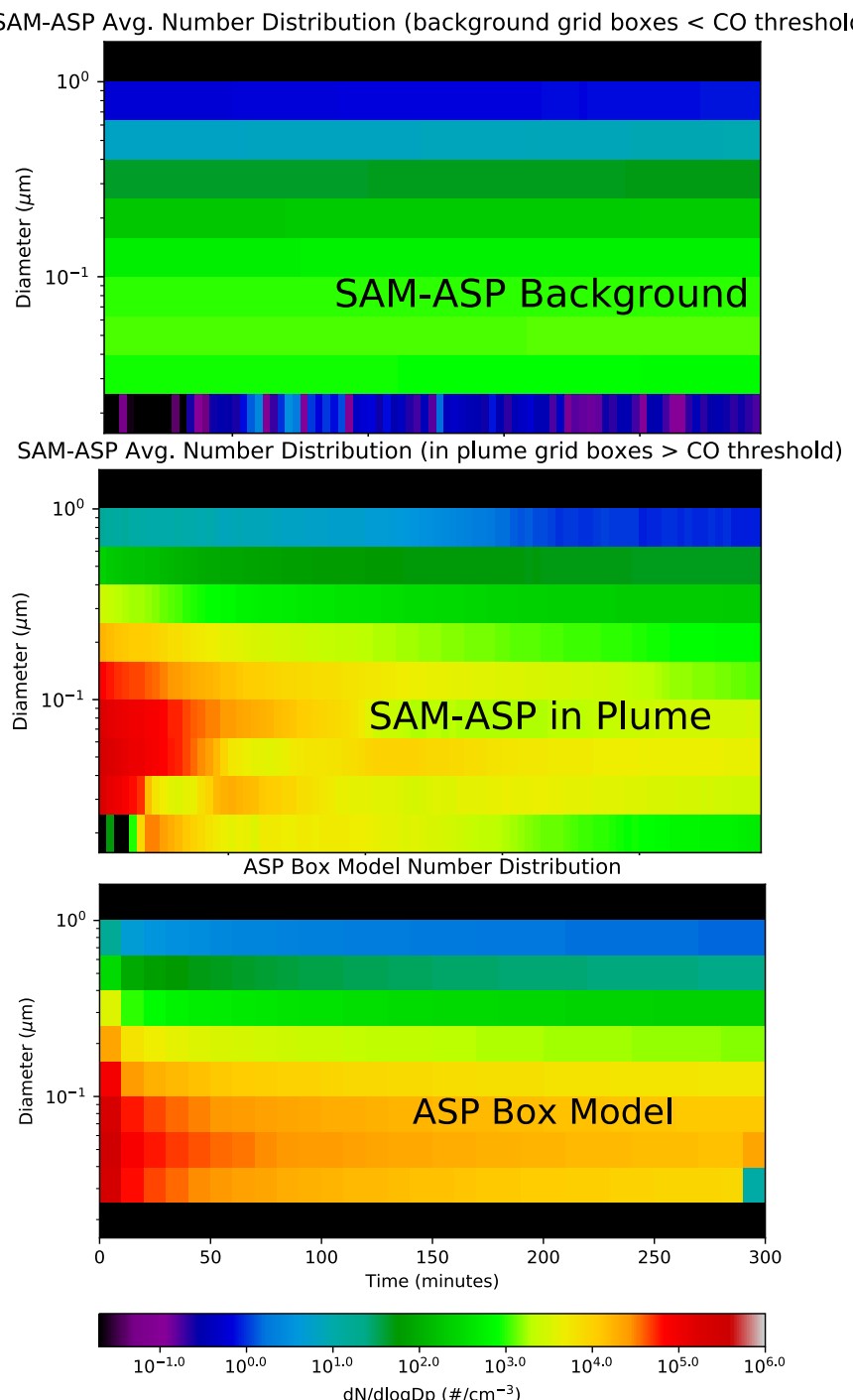

**Figure 6. SAM-ASP (top) and ASP box model (bottom) simulated particle size distribution (dN/dlogD$_p$ cm$^{-3}$) evolution within the Williams fire.**