# Peer review of "Simulating Forest Fire Plume Dispersion, Chemistry, and Aerosol Formation Using SAM-ASP version 1.0"

_Geoscientific Model Development, 2019_

## Referee Comment (RC1) · Anonymous Referee #1 · 26 Sep 2019

Major Comments:

1. Generally I think this paper needs more details. The main point of this journal is to describe the details of the model being used. In most cases the authors simply refereed the readers to another paper to get the specific details. I think most of the subsections is Section 2 Models, could and should be expanded upon. Specifically, it seems there should be more emphasis on the gas-phase chemistry as it seems like a main motivation in the introduction is ozone, yet this subsection is less than 5 sentences long.

2. According to this paper (Yokelson et al. 2013) , it is possible that the NEMR values

depending on the time and location (near the boundary layer vs. free troposphere) might distort the results / meaning of the values of the NEMR, specifically when looking at background $CO_2$. Was this looked at or dealt with? How do the background values of $CO_2$ vary? Also, I would consider mentioning this paper to caution people about the blind use of this technique.

3. It also may be a good idea to compare and contrast the differences between the new model and the box model you used before? This could help the readers further understand Figure 3.

4. For the NEMR technique, why was $CO_2$ background concentrations used only for OA? It seems like there is only primary OA and no secondary OA being produced (in Fig. 4). I would assume the second bump in the measurements is from SOA? Would it be more appropriate to use the change in concentration of CO instead? Or was this looked at and the authors decided $CO_2$ was more appropriate?

Minor comments:

On line 19-20 of p. 4 the authors mention that the hygroscopicity of the aerosol decreases with aging, but I thought the opposite was true, as the aerosol ages, it becomes more hygroscopic.

On the same line you mention this consistent with aerosols from the SAFARI-2000 studies, but isn't that study looking at savannah fires? I'm just wondering even though you aren't looking at these types of fires is still appropriate for your study?

On line 18 of p. 5 you discuss the BC mixing-rule options, I am more just looking for clarification, are all of these mixing states considered, or is just one chosen?

On line 1-3 of p. 6 This is perhaps a more specific example of the first major comment, but it doesn't seem all that helpful to explain that the coupling is similar to another coupling from another paper.

On line 10 of p. 6 The authors mention that the photolysis rates are calculated from a

look up table which are depending on the zenith angle and overheard ozone column but later on in the paper, the authors mention that the photolysis rates are constant. Is this the same thing? The values from the look up tables are all constant?

For Fig. 3 I know the authors explain that something isn't right with the ammonia but did you have any further details on it? That result seems very peculiar. Also for the caption in Fig 3, it would be helpful to the readers to explain the meaning of the horizontal and vertical error bars on the measurements like the authors do in the text.

Line 9 of p. 8, is this really supposed to say Fig. 3?

Pitfalls with the use of enhancement ratios or normalized excess mixing ratios measured in plumes to characterize pollution sources and aging R. J. Yokelson, M. O. Andreae , and S. K. Akagi (2013)

---

## Referee Comment (RC2) · Anonymous Referee #2 · 13 Oct 2019

The manuscript by Lonsdale et al. describes a coupled plume-scale process model that combines the System for Atmospheric Modelling (SAM) and the Aerosol Simulation Program (ASP). Although both SAM and ASP have been developed and extensively used previously, their coupling is a new step. The coupled SAM-ASP model is undoubtedly a useful tool that can help the atmospheric community in studying the near-source smoke plume chemical and physical evolution that cannot be adequately represented in regional and global models. However, I find that the manuscript is too short, with multiple important points not being sufficiently addressed and explained. Furthermore, the presumed advantages of an explicit simulation of the dispersion of a smoke plume compared to a previous single-box model simulation are not convinc-

ingly demonstrated. My specific comments and recommendations for improving this manuscript are provided below.

Specific comments

Introduction: I suggest that the authors better explain the place of their modeling tool among plume models that were available previously. A useful (albeit somewhat outdated) review of such models can be found in Goodrick et al. (2013). I also suggest that anticipated effects of unphysical mixing of biomass burning emissions within grid boxes of 3D CTMs on simulations of air pollution be explained in more detail specifically in the case of particulate matter (based, e.g., on the findings by Bian et al. (2017), Hodshire et al. (2019) and Konovalov et al. (2019)): while the authors gave some idea about these effects in the case of ozone, they did not provide any hints on how unphysical mixing can affect PM simulations.

Sect. 2: I suggest that the title and structure of this section be revised by taking into account that the goal of this manuscript is not to introduce several "models" but rather only one coupled model (SAM-ASP). I suggest specifically that Sect. 2 be entitled as the present Sect 2.3, while Sects. 2.1 and 2.2 that describe the modules (previously developed) of SAM-ASP be merged, and the present Sect. 2.3 be appropriately renamed.

Sect. 2.2: This section is a way too short. It would be helpful if the authors provided information about specific turbulence and cloud parameterizations, a possibility to model aerosol-cloud interactions, limitations associated with basic physical assumptions involved in the model, model grid and typical temporal resolution, an algorithmic language used, etc.

Sect. 2.3: This is, in my understanding, the key section of this paper, and as such, it is also a way too short. In this section, I would expect to find many technical details, such as algorithmic languages used in the model code, a numerical solver, system requirements, availability of parallel computing algorithms, flexibility of the SAM-ASP configuration, etc. I suggest that this section be extended accordingly. Could the authors also explain if the current version of SAM-ASP can be used to simulate aerosol-cloud interactions, if (and how) the wind shear is taken into account in the current Lagrangian configuration of the model, and how the mass emission fluxes can be converted into the initial conditions? I also recommend that Figure 2 from Sakamoto et al. (2016) (to which a reader is referred) be reproduced (possibly with revisions) in this paper.

Sect. 3: Can the authors consider moving the content of this section to Sect. 4?

p. 7, l. 15: Do the authors mean that the emissions are initially distributed evenly between 1200 and 1400 m? If so, could the authors comment on why, according to Fig. 2, the plume is located between 900 and 1300 m after just one hour? Is it initially propagating downward?

P. 7, l. 15, 16: I suggest that the authors explain their choice of the initial horizontal width of the plume. I see that according to Fig. 1, it was about 5 km, while the corresponding scale of the fire (covering 81 hectares) was $\sim 1$ km. Was the fire rectangular?

p. 7, l. 28, 29 and Fig. 2: Can the authors provide NEMR for OA with respect to CO? This will make the results for OA more consistent with the results for the gaseous species and also give to a reader a clue about the OA initial concentration (which determines the OA gas-particle partitioning).

p. 7, l. 7. Can the authors explain how they estimated the age uncertainty?

p. 8, l. 1. Can the authors discuss possible reasons for the underestimation of dispersion in the first two hours of their simulation? Does this bias depend on any options used in the SAM configurations?

p. 8, l. 23-28. The authors found that the behavior of OA at the edges of the plume is different from that near the core. However, I wonder if this difference is important when evaluating the average NEMR across the plume? Fig. 4 seems to suggest that the edge effects could indeed be significant (with respect to the evolution of the average

NEMR), but the firm conclusion is hardly possible as the CO dispersion rates in the box model and SAM-ASP are very different. I suggest therefore that the authors make an additional experiment where the CO dispersion rate in the box model is adjusted to that in SAM-ASP. A positive outcome of such an experiment will make the paper much stronger.

Sect. 5: Conclusions look unusually too concise for a GMD paper and should be considerably extended. It should be made clear, in particular, that when compared to observations, the simulation with SAM-ASP did not show any significant differences with respect to a much simpler box model simulation.

Sect. 6: In my understanding, GMD authors are normally expected to provide free access to their modeling tools. But in this case, the access is to be granted by a person who is not even a co-author of this paper. Can the authors consider providing easier access to their model?

Minor comments

p. 2, l 15: I suggest using "reviewed" instead of "described".

p. 2, l 32: CTMs do not "make" emission estimates but only use them.

p. 2, l. 26: I suggest removing the word "size".

Sect. 2.1.2: I suggest moving the description of the settings specific for the numerical experiments performed with SAM-ASP in this particular study to Sect. 4.

p. 7, l. 33: "PAN, NOx..."=> "NEMRS for PAN, NOx, ..."

Fig. 3: The figure caption should mention that the box model results are adopted from Alvarado et al. (2015) (if this is so).

p. 11, l. 4, Bian H.,..., 2017: Is it the correct reference?

References

Bian, Q., Jathar, S. H., Kodros, J. K., Barsanti, K. C., Hatch, L. E., May, A. A., Kreidenweis, S. M., and Pierce, J. R.: Secondary organic aerosol formation in biomass-burning plumes: theoretical analysis of lab studies and ambient plumes, Atmos. Chem. Phys., 17, 5459–5475, https://doi.org/10.5194/acp-17-5459-2017, 2017.

Goodrick, S. L., Achtemeier, G. L., Larkin, N. K., Liu, Y., and Strand, T. M.: Modelling smoke transport from wildland fires: a review, Int. J. Wildland Fire, 22, 83–94, doi:10.1071/WF11116, 2013.

Hodshire, A. L., Bian, Q., Ramnarine, E., Lonsdale, C. R., Alvarado, M. J., Kreidenweis, S. M., Jathar, S. H., and Pierce, J. R.: More than emissions and chemistry: Fire size, dilution, and background aerosol also greatly influence nearfield biomass burning aerosol aging, J. Geophys. Res.-Atmos., 124, 5589–5611, https://doi.org/10.1029/2018JD029674, 2019.

Konovalov, I. B., Beekmann, M., Golovushkin, N. A., and Andreae, M. O.: Nonlinear behavior of organic aerosol in biomass burning plumes: a microphysical model analysis, Atmos. Chem. Phys., 19, 12091–12119, https://doi.org/10.5194/acp-19-12091-2019, 2019.

---

## Author Comment (AC1) · 11 Jun 2020

We than the reviewers for their careful reading of our manuscript and their helpful comments. We have revised our paper in response to their comments and believe this has made the manuscript stronger. Below, reviewer comments are in bold, our responses are in italics, and text added to the paper is in plain text.

**Reviewer #1**

**Major Comments:**
**1. Generally I think this paper needs more details. The main point of this journal is to describe the details of the model being used. In most cases the authors simply refereed the readers to another paper to get the specific details. I think most of the subsections is Section 2 Models, could and should be expanded upon. Specifically, it seems there should be more emphasis on the gas-phase chemistry as it seems like a main motivation in the introduction is ozone, yet this subsection is less than 5 sentences long.**

*We agree and have substantially added to the text of the paper, as illustrated in our response to comments by Reviewers 1 and 2 below. Specifically, we have added the following text discussing the gas-phase chemistry in ASP.*

P5, L14 to P6, L19: The gas-phase chemistry within ASP v2.1 is described in detail in Alvarado et al. (2015). The chemical mechanism is integrated using a Gear-algorithm type solver. The ASP v2.1 gas phase chemical mechanism includes 1608 reactions between 621 species. All gas-phase chemistry for organic compounds containing 4 carbons or less has been "unlumped," i.e. the chemistry for each individual organic compound is explicitly resolved. This was done by following the reactions of the Leeds Master Chemical Mechanism (MCM) v3.2 (http://mcm.leeds.ac.uk/MCM/, accessed June 2012; Jenkin et al., 1997, 2003; Saunders et al., 2003; Bloss et al., 2005) for these species. The chemical mechanism of isoprene within ASP v2.1 follows the Paulot et al. (2009a,b) isoprene scheme, as implemented in GEOS-Chem and including corrections based on more recent studies (e.g., Crounse et al., 2011, 2012). The (lumped) chemistry for all other organic compounds in ASP v2.1 follows the Regional Atmospheric Chemistry Mechanism (RACM) v2 (Goliff et al., 2013).

Like most organic compounds, semi-volatile organic compounds (SVOCs) will react with OH. Most mechanisms for this chemistry (e.g., Dzepina et al., 2009) parameterize this chemistry by assuming that the SVOCs react with OH to form a lower volatility SVOC, as in the reaction:

$$\text{SVOC}_i + \text{OH} \xrightarrow{k_{OH}} \mu\text{SVOC}_{i-n} \qquad \text{(R1)}$$

where $\mu$ is the relative mass gain due to oxidation (e.g. via O addition), $k_{OH}$ is the reaction rate with OH, and n is the "volatility shift", or by how many factors of 10 to lower the C* of the product with each OH reaction. This simplified chemistry can be

extended to account for the fact that the SVOCs could fragment during oxidation, leading to higher volatility products:

$$\text{SVOC}_i + \text{OH} \xrightarrow{k_{OH}} \mu(1-\alpha)\text{SVOC}_{i-n} + \mu\alpha\text{SVOC}_{i+1} + \alpha\text{VOC}_j \tag{R2}$$

where α is the fraction of SVOC$_i$ that fragments into SVOC$_{i+1}$ and VOC$_j$. Shrivastava et al. (2013) used a similar approach to show that adding SVOC fragmentation to WRF-Chem simulations of the Mexico City Plateau improved the model's ability to simulate the observed concentrations of SOA. However, the highly simplified chemistry of Reactions R1 or R2 is not appropriate for situations where reactions with the SVOC compounds are a potentially significant sink of OH, such as in a concentrated smoke plume. Thus in ASP v2.1, the average, lumped chemistry of the SVOCs is instead parameterized in a more realistic manner for a generic organic species, following the idea of "mechanistic reactivity" (e.g., Seinfeld and Pandis, 1998). After reaction with OH SVOCs produce peroxy radicals (RO$_2$), which can react with NO to form NO$_2$ and HO$_2$, thereby regenerating OH and forming O$_3$. Reactions R3 and R4 show this more general chemical mechanism for the SVOCs:

$$\text{SVOC}_i + \text{OH} \xrightarrow{k_{OH}} \text{RO}_2 \tag{R3}$$

$$\text{RO}_{2,i} + \chi\text{NO} \xrightarrow{k_{RO_{2,i}}} \mu(1-\alpha)\text{SVOC}_{i-n} + \mu\alpha\text{SVOC}_{i+1}$$
$$+ \alpha\text{VOC}_j + \beta\text{NO}_2 + \delta\text{HO}_2 \tag{R4}$$

where $k_{RO_{2,i}}$ is assumed to be 4.0×10$^{-12}$ cm$^3$ molecule$^{-1}$ s$^{-1}$ based on the reaction rate for the peroxy radicals from long-chain alkanes and alkenes with NO in RACM2 (Goliff et al., 2013). We can see that χ-β is the number of NO$_x$ lost (implicitly via the addition of a nitrate group to the product SVOCs), 1-δ is the number of HO$_x$ lost, and β+δ is the number of O$_3$ made per reaction (by subsequent reactions of NO$_2$ and HO$_2$ to generate O$_3$). For example, the values for long-chain alkanes (HC8) in the RACM2 mechanism (Goliff et al., 2013) would be χ = 1, δ = 0.63, and β = 0.74, such that 0.26 NO$_x$ and 0.37 HO$_x$ are lost and 1.37 O$_3$ are formed per reaction. Note that the mechanism of Reactions R3 and R4 is still highly simplified: we assume that reaction of SVOC with OH always produces a RO$_2$ radical, and that the RO$_2$ produced does not react with HO$_2$ or another RO$_2$. Also note that Reactions R3 and R4 represent the average chemistry of the unknown species collectively, and may not apply to any individual species in that mixture. Based on the results of Alvarado et al. (2015), we used an OH reaction rate of 1.0×10$^{-12}$ cm$^3$ molecule$^{-1}$ s$^{-1}$ for Reaction R3, and values of μ = 1.075, α = 0.5, n = 1 χ = 1, δ = 0.6, and β = 0.5 as the defaults in ASP v2.1.

**2. According to this paper (Yokelson et al. 2013) , it is possible that the NEMR values depending on the time and location (near the boundary layer vs. free troposphere) might distort the results / meaning of the values of the NEMR, specifically when looking at background CO2. Was this looked at or dealt with? How do the background values of CO2 vary? Also, I would consider mentioning this paper to caution people about the blind use of this technique.**

*We agree that NEMRs are imprecise indicators, especially for plumes that have traveled far from their original source and may have mixed with different types of background air and thus defining a single background concentration to subtract from the plume concentration is not a realistic approach. In the case of the Williams fire, this is less of a concern as the plume was sampled within 5 hours of emission over a short period of time in similar conditions, and background was estimated separately for each sample.*

*In Akagi et al. (2012) where the Williams fire measurements came from, the authors (who include Yokelson of Yokelson et al. (2013)) state that "we compute the background value from the average of a large number of points at the plume altitude (but outside the plume) and then subtract that background from the values obtained in the plume." They further state that "When comparing NEMR that were determined using data from two different instruments, error can be introduced due to the different time responses of the instruments. However, this error can be largely eliminated for continuous instruments by deriving the NEMR from a comparison of the integrals over the whole plume sample. In addition, in the dispersed, downwind plume, the excess mixing ratios tended to vary slowly in time and space compared to measurement frequency."*

*We have added some of the above discussion, and a general warning about the NEMR technique, to the revised manuscript:*

P10, L26-30: Note that in general, NEMRs are imprecise indicators of chemical changes, especially for plumes that have traveled far from their original source and may have mixed with different types of background air and thus defining a single background concentration to subtract from the plume concentration is not a realistic approach (e.g., Yokelson et al., 2013). However, for the Williams fire, the excess mixing ratios downwind tended to vary slowly in time and space compared to measurement frequency and the background value was computed from the average of a large number of points at the plume altitude (but outside the plume, Akagi et al., 2012).

**3. It also may be a good idea to compare and contrast the differences between the new model and the box model you used before? This could help the readers further understand Figure 3.**

*We agree that we need to make clear that the differences between the models are mainly in how plume mixing and gradients are treated, and have added the below text to the revised manuscript:*

P11, L7-29: For better comparison between the ASP v2.1 box model of Alvarado et al. (2015) and SAM-ASP v1.0, all emission ratios and background concentrations were made identical in box models. The same gas-phase chemical mechanism, aerosol thermodynamics routines and parameters, aerosol size distribution routines and parameters, and other chemical parameters were used. Thus, the key difference between the two models is the treatment of plume dilution and mixing (with minor differences due to vertical temperature, pressure, and humidity variations in SAM-ASP v1.0 versus constant parameters used in ASP v2.1). In ASP v2.1, the plume is a single well-mixed box and dilution is parameterized by assuming a one-way addition of background air to the plume. As in Mason et al. (2001), we assume a Lagrangian parcel of fixed height (H) and length, but variable width y(t). This assumes the plume is well-mixed vertically and capped at the top and bottom by a strong stable layer (or the surface). The temperature and pressure of the parcel are assumed to be constant. The effect of plume dispersion on concentrations is then (Mason et al., 2001; Alvarado, 2008):

$$\left(\frac{dC_q}{dt}\right)_{disp} = -\frac{1}{y(t)}\frac{dy(t)}{dt}\left(C_q - C_q^a\right) \qquad (2)$$

where $C_q$ is the concentration of species q within the parcel (molecules/cm$^3$) and $C_q^a$ is the concentration of species q in the atmosphere outside of the parcel. The form of y(t) is assumed to be y(t) = $y_o^2$ + $8K_y t^2$ , where $y_o$ is the initial plume width (Mason et al., 2001). $K_y$ represents the horizontal diffusivity of the atmosphere. The effect of plume dispersion then becomes

$$\left(\frac{dC_q}{dt}\right)_{disp} = -\frac{4K_y}{(y_o^2 - 8K_y t^2)}\left(C_q - C_q^a\right) \qquad (3)$$

This equation is used with the observations of the rate of change of excess CO in the Williams fire plume to derive best fit values for $K_y$ using the observed value of $y_o$.

In SAM-ASP v1.0, horizontal and vertical mixing between the boxes of the Lagrangian wall are calculated as part of the tracer transport routines of SAM (Khairoutdinov and Randall, 2003). In addition, unlike the ASP v2.1 box model of Alvarado et al. (2015), plume gradients are preserved in SAM-ASP v1.0. Thus, the chemistry taking place in the center of the plume may differ from that in the edges of the plume, potentially changing the plume average NEMRs from those calculated with the well-mixed box assumption in ASP v2.1.

**4. For the NEMR technique, why was CO2 background concentrations used only for OA? It seems like there is only primary OA and no secondary OA being produced (in Fig. 4). I would assume the second bump in the measurements is from SOA? Would it be more appropriate to**

**use the change in concentration of CO instead? Or was this looked at and the authors decided CO2 was more appropriate?**

*In the original observation paper (Akagi et al., 2012), the authors state that "We obtained initial mass emission ratios for the AMS species, rBC, and PM2.5 to CO2 since CO2 was measured on the same inlet." Thus we have preserved that approach in our paper to ensure that we are only ratioing quantities measured in the same sample.*

*We have added this explanation to the revised manuscript:*

P11, L1-3: Note that $\Delta CO_2$ was used to as the NEMR denominator for OA, as in Akagi et al. (2012) and Alvarado et al. (2015), as in the field study OA and $CO_2$ were measured on the same inlet while CO was measured on a different inlet.

*We also note that it isn't the case that "**there is only primary OA and no secondary OA being produced (in Fig. 4)."** What is happening is that the POA is evaporating, and this is being balanced by oxidation of the SVOCs in the gas-phase, which then condenses as SOA. We have tried to clarify this in the revised manuscript:*

P12, L29-30: Thus, in both models the initial POA partially evaporates, but this is balanced by oxidation of the S/IVOCs in the gas-phase, which then condense as SOA.

**Minor comments:**
**On line 19-20 of p. 4 the authors mention that the hygroscopicity of the aerosol de- creases with aging, but I thought the opposite was true, as the aerosol ages, it becomes more hygroscopic.**

*The reviewer is correct than the organic aerosol generally becomes more oxygenated, and thus hygroscopic, with time. However, the study of Magi and Hobbs (2003) was looking at the increase of total aerosol scattering (including inorganic components) with humidification and did report a lower value for the aged smoke (1.4) than for the fresh smoke (1.7-1.79). Also, Engelhart et al. (2012) (https://www.atmos-chem-phys.net/12/7285/2012/acp-12-7285-2012.pdf) showed that several biomass burning fires had lower hygroscopicity for aged smoke than for fresh smoke for fires with a high initial inorganic content. We have revised the text to more precisely state the nature of the Magi and Hobbs (2003) measurement:*

P5, L4-6: They also showed that the aerosol single scattering albedo increased in the first hour of aging from 0.87 to 0.90 and that the change of total aerosol light scattering with humidification decreased with aging, consistent with SAFARI-2000 studies of Magi and Hobbs (2003) and Reid et al. (2005).

**On the same line you mention this consistent with aerosols from the SAFARI-2000 studies, but isn't that study looking at savannah fires? I'm just wondering even though you aren't looking at these types of fires is still appropriate for your study?**

*The Timbavati fire studied in Alvarado and Prinn (2009) was indeed a savannah fire, and thus is not exactly comparable to the chaparral fire examined here. However, the Alvarado and Prinn (2009) paper was the first use of ASP v1.0 iin the peer-reviewed literature, and while the model chemical parameters have changed since then, it is still a relevant historical reference for the model. We have altered the text in this section to make clear than the Alvarado and Prinn study used ASP v1.0, while Alvarado et al. (2015) and this study used ASP v2.1.*

**On line 18 of p. 5 you discuss the BC mixing-rule options, I am more just looking for clarification, are all of these mixing states considered, or is just one chosen?**

*For this study, only the core-shell mixing rule was used. We have added to the text in this section to make this clear:*

P7, L9-10: Only the core-shell parameters were used in this study.

**On line 1-3 of p. 6 This is perhaps a more specific example of the first major comment, but it doesn't seem all that helpful to explain that the coupling is similar to another coupling from another paper.**

*We agree, and have added the following text describing the coupling in more detail:*

P8, L17-30: SAM was updated to transport over 600 gas-phase chemical species calculated in ASP and the 840 aerosol parameters (number concentrations for each bin and mass concentrations for each aerosol species in each bin) and to simulate the emission of the fire smoke by making substantial changes to the tracers.f90 subroutine of SAM. While the number of chemical species and number of size bins is flexible in ASP v2.1 and read in from ASCII input files, these values are hard-coded into the coupled SAM-ASP v1.0 model. There is no coupling of the ASP aerosols with the SAM cloud microphysics scheme in SAM-ASP v1.0.

The tracers.f90 subroutine of SAM was also modified to communicate the solar zenith angle and initialize gas and aerosol tracer concentrations based on SAM meteorological parameters. The coupling takes place via a new ASP subroutine called within tracers.90 in SAM, called SAM_wrapper, which collects the current gas and aerosol concentrations and other parameters and passes them into ASP via the routines in ASP/StepASP.f90. StepASP.f90 performs unit conversions, passes the information into the ASP v2.1 box model, and then calculates the gas-phase chemistry (including heterogeneous chemistry), aerosol thermodynamics, and aerosol coagulation using the routines of ASP

v2.1 described in Section 2.1, which are documented in Alvarado (2008) and Alvarado et al. (2015).

**On line 10 of p. 6 The authors mention that the photolysis rates are calculated from a look up table which are depending on the zenith angle and overheard ozone column but later on in the paper, the authors mention that the photolysis rates are constant. Is this the same thing? The values from the look up tables are all constant?**

*We agree this was unclear – the look-up table changes the photolysis rates with time, but not horizontally or vertically in SAM-ASP. We have clarified this in the text:*

P12, L11-14: The lack of vertical variation in the SAM-ASP plume in Figure 2 may be due to the use of photolysis rates that are not altered by the simulated aerosol scattering and absorption in this version of SAM-ASP. Thus, while the photolysis rates vary with time, they do not vary horizontally or vertically, with future work needed to incorporate in-line, vertically varying photolysis consideration.

**For Fig. 3 I know the authors explain that something isn't right with the ammonia but did you have any further details on it? That result seems very peculiar. Also for the caption in Fig 3, it would be helpful to the readers to explain the meaning of the horizontal and vertical error bars on the measurements like the authors do in the text.**

*We have not further investigated the difference in ammonia. Generally, the results for this gas are very sensitive to the amount of sulfate and nitrate formed in the plume, the dilution of the plume as it affects the volatilization of NH3 from the aerosol, as well as the relative humidity and temperature, all of which slightly differ between ASP v2.1 and SAM-ASP v1.0. We do note that we expect that SAM-ASP v1.0 is providing a more realistic simulation of these parameters, as it includes actual meteorological data in its initialization and better accounts for dilution and the in-plume gradients in sulfate and nitrate. We have added this discussion to the manuscript:*

P12, L8-11: The results for this gas are very sensitive to the amount of sulfate and nitrate formed in the plume, the dilution of the plume as it affects the volatilization of NH3 from the aerosol, as well as the relative humidity and temperature, all of which slightly differ between ASP v2.1 and SAM-ASP v1.0, but we have not yet determined which difference is driving the ammonia discrepancy.

*In addition, the following text has been added to the caption of Figure 3:*
The horizontal error bars indicate the age uncertainty of the measurements while the vertical errors bars are the uncertainty of the measured value.

**Line 9 of p. 8, is this really supposed to say Fig. 3?**

*No, this should have said Figure 2, and has been corrected in the revised manuscript. However, as we have added a new Figure 1, it is again Figure 3,*

**Reviewer #2**

The manuscript by Lonsdale et al. describes a coupled plume-scale process model that combines the System for Atmospheric Modelling (SAM) and the Aerosol Simulation Program (ASP). Although both SAM and ASP have been developed and extensively used previously, their coupling is a new step. The coupled SAM-ASP model is undoubtedly a useful tool that can help the atmospheric community in studying the near-source smoke plume chemical and physical evolution that cannot be adequately represented in regional and global models. However, I find that the manuscript is too short, with multiple important points not being sufficiently addressed and explained. Furthermore, the presumed advantages of an explicit simulation of the dispersion of a smoke plume compared to a previous single-box model simulation are not convincingly demonstrated. My specific comments and recommendations for improving this manuscript are provided below.

*We agree that the example of the Williams fire shown in our manuscript does not convincingly demonstrate that SAM-ASP v1.0 gives substantially better results than the simple box model of ASP v2.1, especially as data on the horizontal and vertical gradients of species in the smoke plume were not available for the Williams fire. However, we chose this fire for this manuscript because the previous ASP v2.1 model result had already been published and peer-reviewed, and so could be used as a for the performance of the coupled SAM-ASP v2.1 model. We plan in future work to apply the SAM-ASP model to the results of recent fire field campaigns (such as WE-CAN) which do have this gradient data, but have submitted this manuscript to GMD to have the baseline model documented and reviewed before performing and publishing those additional studies.*

**Specific comments**
**Introduction: I suggest that the authors better explain the place of their modeling tool among plume models that were available previously. A useful (albeit somewhat out-dated) review of such models can be found in Goodrick et al. (2013).**

*We agree, and have added the following text to the introduction:*

*P3, L32-36: Several types of models have been used to simulate the dispersion and transport of smoke plumes, including box models, Gaussian plume models, Lagrangian puff and particle dispersion models (e.g., CALPUFF, SCIPUFF, HYSPLIT, FLEXPART), and 3D Eulerian models (e.g., Goodrick et al., 2013 and the references therein). A smaller number of models have included the gas (e.g., Mason et al., 2001) and aerosol (e.g., Trentmann et al., 2003) chemistry of these plumes, and a smaller number still have tried to predict how the aerosol size distribution changes within the smoke plume (e.g., Sakamoto et al., 2016; Hodshire et al., 2019b).*

**I also suggest that anticipated effects of unphysical mixing of biomass burning emissions within grid boxes of 3D CTMs on simulations of air pollution be explained in more detail**

**specifically in the case of particulate matter (based, e.g., on the findings by Bian et al. (2017), Hodshire et al. (2019) and Konovalov et al. (2019)): while the authors gave some idea about these effects in the case of ozone, they did not provide any hints on how unphysical mixing can affect PM simulations.**

*We agree, and have added the following text to the introduction:*

P3, L5-15: Similarly, the unphysical mixing of biomass burning emissions into large-scale grid boxes can lead to incorrect estimates of OA concentrations and the aerosol size distribution (e.g., Alvarado et al., 2009; Sakamoto et al., 2016; Bian et al., 2017; Ramnarine et al., 2019; Hodshire et al., 2019b; Konovalov et al., 2019). The net change of OA mass in a smoke plume as it dilutes and ages is determined from the balance between initial emissions, secondary organic aerosol (SOA) production, and evaporation of both primary organic aerosol (POA) and SOA (Bian et al., 2017; Hodshire et al., 2019b). Unphysically diluting biomass burning emissions leads to unphysical evaporation of the POA, reduced the rates of chemical SOA formation, and more of the formed SOA remining in the gas phase in the 3D Eulerian CTMs. Similarly, the unphysical dilution reduces the aerosol number concentration, reducing coagulation rates (Sakamoto et al., 2016; Ramnarine et al., 2019), while the more dilute smoke will not reach the high concentrations needed to nucleate new particles. As the evolution of the aerosol size distribution in smoke plumes is primarily controlled by OA mass changes, coagulation, and nucleation, 3D Eulerian CTMs will have difficulty accurately simulation the aerosol size distribution changes without parameterizing these sub-grid scale processes.

**Sect. 2: I suggest that the title and structure of this section be revised by taking into account that the goal of this manuscript is not to introduce several "models" but rather only one coupled model (SAM-ASP). I suggest specifically that Sect. 2 be entitled as the present Sect 2.3, while Sects. 2.1 and 2.2 that describe the modules (previously developed) of SAM-ASP be merged, and the present Sect. 2.3 be appropriately renamed.**

*We renamed Section 2 "SAM-ASP 2D Lagrangian Model" as requested and renamed Section 2.3 as "Model Coupling." We felt that combining sections 2.1 and 2.2 made the text too confusing when the transition from discussing ASP and SAM occurred, and so have chosen not to change these headings except to add the version numbers for clarity.*

**Sect. 2.2: This section is a way too short. It would be helpful if the authors provided information about specific turbulence and cloud parameterizations, a possibility to model aerosol-cloud interactions, limitations associated with basic physical assumptions involved in the model, model grid and typical temporal resolution, an algorithmic language used, etc.**

*We have greatly expanded this section as requested, see below:*

P7, L26 to P8, L9: The SAM v6.10.10 model is a Fortran code has been used to study aerosol-cloud-precipitation interactions in stratiform and convective clouds (Ovchinnikov et al., 2014; Fan et al., 2009). The standard SAM model (Khairoutdinov and Randall, 2003, http://rossby.msrc.sunysb.edu/~marat/SAM.html) includes different options of detailed cloud microphysics, as well as coupled radiation and land-surface models. SAM is able to resolve boundary layer eddies, while parameterizing smaller-scale turbulence and microphysics for the LES (vs cloud-resolving) model option. The dynamical framework of the model is based on the large eddy simulation (LES) model of Khairoutdinov and Kogan (1999). Besides using the anelastic equations of motion in place of the Boussinesq equations of the LES version, SAM uses a different set of prognostic thermodynamic variables and employs a different microphysics scheme. The computer code was designed to run efficiently on parallel computers using the Message Passing Interface (MPI) protocol. The detailed description of the model equations is given in the appendix A of Khairoutdinov and Randall (2003).

The prognostic thermodynamical variables of the model are the liquid water/ice moist static energy, total nonprecipitating water (vapor + cloud water + cloud ice), and total precipitating water (rain + snow + graupel). The liquid water/ice moist static energy is, by definition, conserved during the moist adiabatic processes including the freezing/melting of precipitation. The cloud condensate (cloud water + cloud ice) is diagnosed using the so-called "all-or-nothing" approach, so that no supersaturation of water vapor is allowed. Despite being called a nonprecipitating water substance, the cloud ice is actually allowed to have a nonnegligible terminal velocity. The partitioning of the diagnosed cloud condensate and the total precipitating water into the hydrometeor mixing ratios is done on every time step as a function of temperature. The diagnosed hydrometeor mixing ratios are then used to compute the water sedimentation and hydrometeor conversion rates.

The finite-difference representation of the model equations uses a fully staggered Arakawa C-type grid with stretched vertical and uniform horizontal grids. The advection of momentum is computed with the second-order finite differences in the flux form with kinetic energy conservation. The equations of motion are integrated using the third-order Adams–Bashforth scheme with a variable time step. All prognostic scalars, including the chemical tracers of ASP v2.1, are advected using a fully three-dimensional positive definite and monotonic scheme of Smolarkiewicz and Grabowski (1990). The subgrid-scale model employs the so-called 1.5-order closure based on a prognostic subgrid-scale turbulent kinetic energy. The model uses periodic lateral boundaries, and a rigid lid at the top of the domain. To reduce gravity wave reflection and buildup, the Newtonian damping is applied to all prognostic variables in the upper third of the model domain. The surface fluxes are computed using Monin–Obukhov similarity. SAM can be driven by reanalysis data that includes large-scale forcings, initial sounding profile, radiation heating rates, and surface fluxes. SAM has the ability to add a large amount of modeled tracer species to the cloud resolving model simulation but does not contain aerosol and chemistry packages.

The SAM model is flexible with different choices for advection scheme, turbulence parameterization, radiation, and cloud microphysics. The configuration used in SAM-ASP v2.1 includes the use of a positive definite monotonic advection scheme with a non-oscillatory option, the 1.5-order TKE closure for sub-grid scale turbulence, the microphysics scheme of Morrison et al. (2005), and the CAM radiation code.

P8, L25-26: There is no coupling of the ASP aerosols with the SAM cloud microphysics scheme in SAM-ASP v1.0.

**Sect. 2.3: This is, in my understanding, the key section of this paper, and as such, it is also a way too short. In this section, I would expect to find many technical details, such as algorithmic languages used in the model code, a numerical solver, system requirements, availability of parallel computing algorithms, flexibility of the SAM-ASP configuration, etc. I suggest that this section be extended accordingly. Could the authors also explain if the current version of SAM-ASP can be used to simulate aerosol-cloud interactions, if (and how) the wind shear is taken into account in the current Lagrangian configuration of the model, and how the mass emission fluxes can be converted into the initial conditions? I also recommend that Figure 2 from Sakamoto et al. (2016) (to which a reader is referred) be reproduced (possibly with revisions) in this paper.**

*We have greatly extended this section as requested. SAM-ASP is a Fortran code. The numerical solvers used are the same as those for SAM and ASP – the main change was to increase the number of tracers in SAM and to call ASP as a subroutine within SAM every time the tracers.f90 routine of SAM is called. The code is as parallel as the original SAM model and was run on 12 processors with 4 GB of RAM each for this study.*

*SAM-ASP v1.0 does not link the aerosol predictions of ASP with the Morrison et al. (2005) microphysics scheme in SAM and so cannot be used for aerosol-cloud interaction studies. Wind shear in the NARR dataset used for boundary conditions also impacts the coupled model – the downwind (x) direction is determined once and from then on the dynamics occur in this 2D plane based on the boundary condition forcing and the model advection and turbulence schemes. As requested, we have added a description of how the mass emission fluxes are converted into initial plume concentrations in this section. We have also reproduced the figure of Sakamoto et al. (2016) as requested.*

[revised manuscript text omitted]

When ASP v2.1 is run as a Lagrangian box model, it needs the initial concentrations within the plume to be specified. However, as SAM-ASP v1.0 can simulate the dispersion of the smoke horizontally and vertically, we added the capability to calculate the initial concentrations based on the mass emissions flux (M, kg burned m$^{-2}$ s$^{-1}$), emission factors (EF, g (kg burned)$^{-1}$), and fire area (A, m$^2$ and assumes a square shape) for biomass burning species (Akagi et al., 2011; Sakamoto et al., 2015). The formula is:

$$\Delta m_q = M \cdot EF_q \cdot A \cdot \Delta t / BM \qquad (1)$$

Where $\Delta m_q$ is the mass mixing ratio (kg q/kg air) of species q, which are the units used in SAM for tracer species, and BM is the mass of air in the emission box (in kg). This allows SAM-ASP v1.0 to better represent a wide range of fire sizes and intensities. To reduce computation time, ASP is only called in the boxes that are impacted by smoke in each SAM timestep, defined as any grid box having a concentration of CO greater than a user-defined threshold (based on background concentrations determined by ambient fire measurements, here 150 ppb). The coupled SAM-ASP v1.0 model was run on 12 processors with 4 GB each, which should be considered the minimum system requirements.

**Sect. 3: Can the authors consider moving the content of this section to Sect. 4?**

*We have moved this section as requested.*

**p. 7, l. 15: Do the authors mean that the emissions are initially distributed evenly between 1200 and 1400 m? If so, could the authors comment on why, according to Fig. 2, the plume is located between 900 and 1300 m after just one hour? Is it initially propagating downward?**

*We note that we were incorrect in our initial paper, the plume was injected between 1200 m and 1360 m with the emissions distributed proportional to the mass of air in the box (i.e., the air density), and thus were not evenly distributed. This, combined with the vertical and horizontal wind shear present in our background wind field, led to the downward propagation observed in Figure 1 (now Figure 2). We have clarified this in the text:*

P10, L34-37: Note the initial plume was distributed across two horizontal gird boxes (initial plume width of 1 km) and four vertical grid boxes (initial height from 1200 m to 1360 m) and was rectangular. The emissions were distributed proportional to the

density of air in each gridbox, and initially propagated downward due to wind shear and diffusion.

**P. 7, l. 15, 16: I suggest that the authors explain their choice of the initial horizontal width of the plume. I see that according to Fig. 1, it was about 5 km, while the corresponding scale of the fire (covering 81 hectares) was ~ 1 km. Was the fire rectangular?**

*The smoke was initially emitted into two horizontal grid boxes, for an initial plume width of 1 km, consistent with the value assumed for the box model of Alvarado et al. (2015). The plume rapidly expanded due to dilution giving the larger 5 km scale in the Figure 1. We have clarified this in the text as noted in the above comment:*

P10, L34-35: Note the initial plume was distributed across two horizontal gird boxes (initial plume width of 1 km) and four vertical grid boxes (initial height from 1200 m to 1360 m) and was rectangular.

**p. 7, l. 28, 29 and Fig. 2: Can the authors provide NEMR for OA with respect to CO? This will make the results for OA more consistent with the results for the gaseous species and also give to a reader a clue about the OA initial concentration (which determines the OA gas-particle partitioning).**

*As noted for Reviewer #1 above, in the original observation paper (Akagi et al., 2012), the authors state that "We obtained initial mass emission ratios for the AMS species, rBC, and PM2.5 to CO2 since CO2 was measured on the same inlet." This we have preserved that approach in our paper to ensure that we are only ratioing quantities measured in the same sample. We have added this explanation to the revised manuscript:*

P11, L1-3: Note that $\Delta CO_2$ was used to as the NEMR denominator for OA, as in Akagi et al. (2012) and Alvarado et al. (2015), as in the field study OA and $CO_2$ were measured on the same inlet while CO was measured on a different inlet.

**p. 7, l. 7. Can the authors explain how they estimated the age uncertainty?**

*Age uncertainty was estimated the same way as in Akagi et al. (2012), where the one-sigma uncertainty in the average horizontal wind speeds during the sampling period were propagated through the plume age calculation, assuming the distance calculation was accurate. We have clarified this in the text:*

P11, L3-5: The uncertainty in the Lagrangian age was calculated as in Akagi et al. (2012), where the one-sigma uncertainty in the average horizontal wind speeds during the sampling period were propagated through the plume age calculation, assuming the distance calculation was accurate.

**p. 8, l. 1. Can the authors discuss possible reasons for the underestimation of dispersion in the first two hours of their simulation? Does this bias depend on any options used in the SAM configurations?**

> *The underestimation in dispersion in the first two hours may indeed be due to the settings used in our SAM configuration and should be explored in future work when staff and funding is available. For now, we have added the following text to our manuscript:*

> P11, L39 to P12, L1: As ASP v2.1 currently uses a fixed function to simulate dilution, we were unable to test how using the SAM-ASP predicted dilution of CO to ASP v2.1 would alter the box model results.

**p. 8, l. 23-28. The authors found that the behavior of OA at the edges of the plume is different from that near the core. However, I wonder if this difference is important when evaluating the average NEMR across the plume? Fig. 4 seems to suggest that the edge effects could indeed be significant (with respect to the evolution of the average NEMR), but the firm conclusion is hardly possible as the CO dispersion rates in the box model and SAM-ASP are very different. I suggest therefore that the authors make an additional experiment where the CO dispersion rate in the box model is adjusted to that in SAM-ASP. A positive outcome of such an experiment will make the paper much stronger.**

> *We agree that an additional experiment where the CO dispersion rate in the box model is constrained to that predicted by SAM-ASP would allow us to separate the effects of plume gradients from that of the dilution changes. However, right now ASP v2.1 can only perform dilution according to the parameterization of Mason et al. (2001) (see P11, L13-24 of the revised paper), and thus such a study would be a significant undertaking that we are not able to perform at this time.*

**Sect. 5: Conclusions look unusually too concise for a GMD paper and should be considerably extended. It should be made clear, in particular, that when compared to observations, the simulation with SAM-ASP did not show any significant differences with respect to a much simpler box model simulation.**

> *We have added to the Conclusions section as shown below:*

> P13, L3-7: We have described a new coupled model, SAM-ASP v1.0, for simulating the gas and aerosol chemistry within biomass burning smoke plumes. The model adds the Aerosol Simulation Program v2.1 (ASP v2.1) as an embedded subroutine within the System for Atmospheric Modeling v6.10 (SAM v6.10). When configured as a 2D Lagrangian wall, the newly coupled SAM-ASP model allows for a detailed examination of the chemical and physical evolution of fine-scale biomass burning plumes

P13, L13-16: However, when compared to observations, the simulation with SAM-ASP did not show any significant differences with respect to a much simpler box model simulation, potentially because the photolysis rates within both simulations were identical, rather than allowing the photolysis rates to vary with predicted aerosol concentrations.

**Sect. 6: In my understanding, GMD authors are normally expected to provide free access to their modeling tools. But in this case, the access is to be granted by a person who is not even a co-author of this paper. Can the authors consider providing easier access to their model?**

*Prior to submitting this paper to GMD, we discussed with the System for Atmospheric Modeling (SAM) model developer, Dr. Marat Khairoutdinov, our desire to include the entire SAM-ASP model code in a zenodo repository in order to meet the GMD requirements. He expressed concerns since this would violate the SAM user agreement. As a compromise he offered to host the SAM-ASP code in its entirety on the existing SAM website, which allows one to freely download the code after requesting a username and password. We have hopefully worded the accessibility clearly in the Data Availability section. Additionally, we have added the ASPv2.1 source code separately to the zenodo repository (providing an updated doi), since this portion of the model code is open source.*

**Minor comments**
**p. 2, l 15: I suggest using "reviewed" instead of "described".**

*Fixed as requested.*

**p. 2, l 32: CTMs do not "make" emission estimates but only use them.**

*This was a typo, fixed to "take".*

**p. 2, l. 26: I suggest removing the word "size".**

*Assuming this is P2 L36, we have deleted the word "size" as requested to say "aerosol evolution".*

**Sect. 2.1.2: I suggest moving the description of the settings specific for the numerical experiments performed with SAM-ASP in this particular study to Sect. 4.**

*We have moved the text as requested.*

**p. 7, l. 33: "PAN, NOx. . ."=> "NEMRS for PAN, NOx, . . ."**

*Fixed as requested.*

**Fig. 3: The figure caption should mention that the box model results are adopted from Alvarado et al. (2015) (if this is so).**

*The caption now reads* "Figure 4. Cross-plume averaged ΔCO and $O_3$, PAN, $NO_x$, HONO and $NH_3$ NEMRs (ΔX/ΔCO) as a function of plume age for the ASP box model (solid line, reproduced from Alvarado et al., 2015) and SAM-ASP model (dashed-line) results compared to measurements from Akagi et al. (2012) (dots)."

**p. 11, l. 4, Bian H.,. . ., 2017: Is it the correct reference?**

*We apologize, the reference should be:*
Bian, Q., Jathar, S. H., Kodros, J. K., Barsanti, K. C., Hatch, L. E., May, A. A., Kreidenweis, S. M., and Pierce, J. R.: Secondary organic aerosol formation in biomass-burning plumes: theoretical analysis of lab studies and ambient plumes, Atmos. Chem. Phys., 17, 5459–5475, https://doi.org/10.5194/acp-17-5459-2017, 2017.
*We have corrected this in the revised manuscript.*